# Learning on Large Graphs
# using Intersecting Communities

**Ben Finkelshtein**
University of Oxford

**İsmail İlkan Ceylan**
University of Oxford

**Michael Bronstein**
University of Oxford / AITHYRA

**Ron Levie**
Technion –- Israel Institute of Technology

## Abstract

Message Passing Neural Networks (MPNNs) are a staple of graph machine learning. MPNNs iteratively update each node's representation in an input graph by aggregating messages from the node's neighbors, which necessitates a memory complexity of the order of the *number of graph edges*. This complexity might quickly become prohibitive for large graphs provided they are not very sparse. In this paper, we propose a novel approach to alleviate this problem by approximating the input graph as an intersecting community graph (ICG) – a combination of intersecting cliques. The key insight is that the number of communities required to approximate a graph *does not depend on the graph size*. We develop a new constructive version of the Weak Graph Regularity Lemma to efficiently construct an approximating ICG for any input graph. We then devise an efficient graph learning algorithm operating directly on ICG in linear memory and time with respect to the *number of nodes* (rather than edges). This offers a new and fundamentally different pipeline for learning on very large non-sparse graphs, whose applicability is demonstrated empirically on node classification tasks and spatio-temporal data processing.

## 1   Introduction

The vast majority of graph neural networks (GNNs) learn representations of graphs based on the message passing paradigm [22], where every node representation is synchronously updated based on an aggregate of messages flowing from its neighbors. This mode of operation hence assumes that the set of all graph edges are loaded in the memory, which leads to a memory complexity bottleneck. Specifically, considering a graph with $N$ nodes and $E$ edges, message-passing leads to a memory complexity that is *linear in the number of edges*, which quickly becomes prohibitive for large graphs especially when $E \gg N$. This limitation motivated a body of work with the ultimate goal of making the graph machine learning pipeline amenable to large graphs [24, 12, 13, 7, 65].

In this work, we take a drastically different approach to alleviate this problem and design a novel graph machine learning pipeline for learning over large graphs with memory complexity *linear in the number of nodes*. At the heart of our approach lies a graph approximation result which informally states the following: every undirected graph with node features can be represented by a linear combination of intersecting communities (i.e., cliques) such that the number of communities required to achieve a certain approximation error is *independent of the graph size* for dense graphs. Intuitively, this results allows us to utilize an "approximating graph", which we call *intersecting community graph* (ICG), for learning over large graphs. Based on this insight — radically departing from the standard message-passing paradigm — we propose a new class of neural networks acting on the ICG and on the set of nodes from the original input graph, which leads to an algorithm with memory and runtime complexity that is *linear in the number nodes*.

38th Conference on Neural Information Processing Systems (NeurIPS 2024).

Our analysis breaks the traditional dichotomy of approaches that are appropriate either for small and dense graphs or for large and sparse graphs. To our knowledge, we present the first rigorously motivated approach allowing to efficiently handle graphs that are both large and dense. The main advantage lies in being able to process a very large non-sparse graph without excessive memory and runtime requirements. The computation of ICG requires linear time in the number of edges, but this is an offline pre-computation that needs to be performed *once*. After constructing the ICG, learning to solve the task on the graph — the most computation demanding aspect — is very efficient, since the architecture search and hyper-parameter optimization is linear in the number of nodes.

**Context.** The focus of our study is *graph-signals*: undirected graphs with node features, where the underlying graph is given and fixed. This is arguably the most common learning setup for large graphs, including tasks such as semi-supervised (transductive) node classification. As we allow varying features, our approach is also suitable for spatiotemporal tasks on graphs, where a single graph is given as a fixed domain on which a dynamic process manifests as time-varying node features. Overall, our approach paves the way for carrying out many important learning tasks on very large and relatively dense graphs such as social networks that typically have $10^8 \sim 10^9$ nodes and average node degree of $10^2 \sim 10^3$ [19] and do not fit into GPU memory.

**Challenges and techniques.** There are two fundamental technical challenges to achieve the desired linear complexity, which we discuss and elaborate next.

*How to construct the* ICG *efficiently with formal approximation guarantees?* Our answer to this question rests on an adaptation of the Weak Regularity Lemma [20, 38] which proves the existence of an approximating ICG in *cut metric* – a well-known graph similarity meausre[1]. Differently from the Weak Regularity Lemma, however, we need to *construct* an approximating ICG. Directly minimizing the cut metric is numerically unstable and hard to solve[2]. We address this in Theorem 3.1, by proving that optimizing the error in Frobenius norm, a much easier optimization target, guarantees a small cut metric error. Hence, we can approximate uniformly "granular" adjacency matrices by low-rank ICGs in cut metric. Uniformity here means that the number of communities $K$ of the ICG required for the error tolerance $\epsilon$ is $K = \epsilon^{-2}$, independently of on any property of the graph, not even its size $N$.[3] This result allows us to design an efficient and numerically stable algorithm. Figure 1 shows an adjacency matrix with an inherent intersecting community structure, and our approximated ICG, which captures the statistics of edge densities, ignoring their granularity.

*How to effectively use* ICG *for efficient graph learning?* We present a signal processing framework and deep neural networks based on the ICG representation (ICG-NN). As opposed to message-passing networks that require $\mathcal{O}(E)$ memory and time complexity, ICG-NNs operate in only $\mathcal{O}(N)$ complexity. They involve node-wise operations to allow processing the small-scale granular behavior of the node features, and community operations that account for the large-scale structure of the graph. Notably, since communities have large supports, ICG-NN can propagate information between far-apart nodes in a single layer.

**Contributions.** Our contributions can be summarized as follows:

- We present a novel graph machine learning approach for large and non-sparse graphs which enables an $\mathcal{O}(N)$ learning algorithm on ICG with one-off $\mathcal{O}(E)$ pre-processing cost for constructing ICG.
- Technically, we prove a constructive version of Weak Regularity Lemma in Theorem 3.1 to efficiently obtain ICGs, which could be of independent theoretical interest.
- We introduce ICG neural networks as an instance of a signal processing framework, which applies neural networks on nodes and on ICG representations .
- Empirically, we validate our theoretical findings on semi-supervised node classification and on spatio-temporal graph tasks, illustrating the scalability of our approach, which is further supported by competitive performance against strong baseline models.

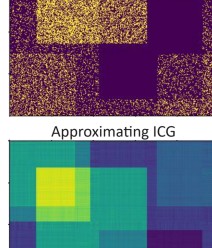

Simple graph

Approximating ICG

Figure 1: **Top**: adjacency matrix of a simple graph. **Bottom**: approximating 5 community ICG.

---

[1][20] proposed an algorithm for finding the ICG, but it has impractical exponential runtime (see Section 7).

[2]The definition of cut metric involves maximization, so minimizing cut metric is an unstable min-max.

[3]Note that the optimal Frobenius error is not uniformly small, since adjacency matrices of simple graphs are $\{0, 1\}$-valued, typically very granular, so they cannot be approximated well by low-rank $\mathbb{R}$-valued ICG matrices. Nevertheless, if we find an approximate Frobenius minimizer, we are guaranteed a small cut metric error.

## 2 A primer on graph-signals and the cut-metric

**Graph signals.** We are interested in undirected (weighted or simple), featured graphs $G = (\mathcal{N}, \mathcal{E}, \mathbf{S})$, where $\mathcal{N}$ denotes the set of nodes, $\mathcal{E}$ denotes the set of edges, and $\mathbf{S}$ denotes the node feature matrix, also called the signal. We will write $N$ to represent the *number of nodes* and $E$ to represent the *number of edges*. For ease of notation, we will assume that the nodes are indexed by $[N] = \{1, \ldots, N\}$. Using the graph signal processing convention, we represent the graphs in terms of a *graph-signal* $G = (\mathbf{A}, \mathbf{S})$, where $\mathbf{A} = (a_{i,j})_{i,j=1}^N \in \mathbb{R}^{N \times N}$ is the adjacency matrix, and $\mathbf{S} = (s_{i,j})_{i \in [N], j \in [d]} \in \mathbb{R}^{N \times D}$ is the $D$-channel signal. We assume that the graph is undirected, meaning that $\mathbf{A}$ is symmetric. The graph may be unweighted (with $a_{i,j} \in \{0,1\}$) or weighted (with $a_{i,j} \in [0,1]$). We additionally denote by $\mathbf{s}_n^\top$ the $n$-th row of $\mathbf{S}$, and denote $\mathbf{S} = (\mathbf{s}_n)_{n=1}^N$. By abuse of notation, we also treat a signal $\mathbf{S}$ as the function $\mathbf{S} : [N] \to \mathbb{R}^D$ with $\mathbf{S}(n) = \mathbf{s}_n$. We suppose that all signals are standardized to have values in $[0,1]$. In general, any matrix (or vector) and its entries are denoted by the same letter, and vectors are seen as columns. The $i$-th row of the matrix $\mathbf{A}$ is denoted by $\mathbf{A}_{i,:}$, and the $j$-th column by $\mathbf{A}_{:,j}$. The Frobenius norm of a square matrix $\mathbf{M} \in \mathbb{R}^{N \times N}$ is defined by $\|\mathbf{M}\|_\mathrm{F} = \sqrt{\frac{1}{N^2} \sum_{n,m=1}^N |b_{n,m}|^2}$, and for a signal $\mathbf{S} \in \mathbb{R}^{N \times D}$ by $\|\mathbf{S}\|_\mathrm{F} = \sqrt{\frac{1}{N} \sum_{n=1}^N \sum_{j=1}^D |s_{n,j}|^2}$. The Frobenius norm of a matrix-signal, with weights $a, b > 0$, is defined to be $\|(\mathbf{M}, \mathbf{S})\|_\mathrm{F} := \sqrt{a \|\mathbf{M}\|_\mathrm{F}^2 + b \|\mathbf{S}\|_\mathrm{F}^2}$. We define the *degree* of a weighted graph as $\deg(\mathbf{A}) := N^2 \|\mathbf{A}\|_\mathrm{F}^2$. In the special case of an unweighted graph we get $\deg(\mathbf{A}) = E$. The pseudoinverse of a full rank matrix $\mathbf{M} \in \mathbb{R}^{N \times K}$, where $N \geq K$, is $\mathbf{M}^\dagger = (\mathbf{M}^\top \mathbf{M})^{-1} \mathbf{M}^\top$.

**Cut-metric.** The *cut metric* is a graph similarity, based on the *cut norm*. We show here the definitions for graphs of the same size; the extension for arbitrary pairs of graphs is based on graphons (see Appendix B.2). The definition of matrix cut norm is well-known, and we normalize it by a parameter $E \in \mathbb{N}$ that indicates the characteristic number of edges of the graphs of interest, so cut norm remains within a standard range. The definition of cut norm of signals is taken from [33].

**Definition 2.1.** *The* matrix cut norm *of $\mathbf{M} \in \mathbb{R}^{N \times N}$, normalized by $E$, is defined to be*

$$\|\mathbf{M}\|_\square = \|\mathbf{M}\|_{\square;N,E} := \frac{1}{E} \sup_{\mathcal{U}, \mathcal{V} \subset [N]} \left| \sum_{i \in \mathcal{U}} \sum_{j \in \mathcal{V}} m_{i,j} \right|. \tag{1}$$

*The* signal cut norm *of $\mathbf{Z} \in \mathbb{R}^{N \times D}$ is defined to be*

$$\|\mathbf{Z}\|_\square = \|\mathbf{Z}\|_{\square;N} := \frac{1}{DN} \sum_{j=1}^D \sup_{w \subset [N]} \left| \sum_{i \in w} z_{i,j} \right|. \tag{2}$$

*The* matrix-signal *cut norm of $(\mathbf{M}, \mathbf{Z})$, with weights $\alpha > 0$ and $\beta > 0$ s.t. $\alpha + \beta = 1$, is defined to be*

$$\|(\mathbf{M}, \mathbf{Z})\|_\square = \|(\mathbf{M}, \mathbf{Z})\|_{\square;N,E} := \alpha \|\mathbf{M}\|_{\square;N,E} + \beta \|\mathbf{Z}\|_{\square;N}. \tag{3}$$

Given two graphs $\mathbf{A}, \mathbf{A}'$, their distance in *cut metric* is the *cut norm* of their difference, namely

$$\|\mathbf{A} - \mathbf{A}'\|_\square = \frac{1}{E} \sup_{\mathcal{U}, \mathcal{V} \subset [N]} \left| \sum_{i \in \mathcal{U}} \sum_{j \in \mathcal{V}} (a_{i,j} - a'_{i,j}) \right|. \tag{4}$$

The right-hand-side of (4) is interpreted as the difference between the edge densities of $\mathbf{A}$ and $\mathbf{A}'$ on the block on which these densities are the most dissimilar. Hence, cut-metric has a probabilistic interpretation. Note that for simple graphs, $\mathbf{A} - \mathbf{A}'$ is a granular function: it has many jumps between the values $-1$, $0$ and $1$. The fact that the absolute value in (4) is outside the integral, as opposed to being inside like in $\ell_1$ norm, means that cut metric has an averaging effect, which mitigates the granularity of $\mathbf{A} - \mathbf{A}'$. The *graph-signal cut metric* is similarly defined to be $\|(\mathbf{A}, \mathbf{S}) - (\mathbf{A}', \mathbf{S}')\|_\square$.

**Cut metric in graph machine learning.** The cut metric has the following interpretation: two (deterministic) graphs are close to each other in cut metric iff they look like random graphs sampled from the same stochastic block model. The interpretation has a precise formulation in view of the Weak Regularity Lemma [20, 38] discussed below. This makes the cut metric a natural notion of similarity for graph machine learning, where real-life graphs are noisy, and can describe the same underlying phenomenon even if they have different sizes and topologies. Moreover, [33] showed that

GNNs with normalized sum aggregation cannot separate graph-signals that are close to each other in cut metric. In fact, cut metric can separate any non-isomorphic graphons [37], so it has sufficient discriminative power to serve as a graph similarity measure in graph machine learning problems. As opposed to past works that used the cut norm to derive theory for existing GNNs, e.g., [46, 40, 33], we use cut norm to derive new methods for a class of problems of interest. Namely, we introduce a new theorem about cut norm – the constructive weak regularity lemma (Theorem 3.1) – and use it to build new algorithms on large non-sparse graphs.

# 3 Approximation of graphs by intersecting communities

## 3.1 Intersecting community graphs

Given a graph $G$ with $N$ nodes, for any subset of nodes $\mathcal{U} \subset [N]$, denote by $\mathbb{1}_{\mathcal{U}}$ its indicator (i.e., $\mathbb{1}_{\mathcal{U}}(i) = 1$ if $i \in \mathcal{U}$ and zero otherwise). We define an *intersecting community graph-signal (*ICG*)* with $K$ classes ($K$-ICG) as a low rank graph-signal $(\boldsymbol{C}, \boldsymbol{P})$ with adjacency matrix and signals given respectively by

$$\boldsymbol{C} = \sum_{j=1}^{K} r_j \mathbb{1}_{\mathcal{U}_j} \mathbb{1}_{\mathcal{U}_j}^{\top}, \qquad \boldsymbol{P} = \sum_{j=1}^{K} \mathbb{1}_{\mathcal{U}_j} \boldsymbol{f}_j^{\top} \tag{5}$$

where $r_j \in \mathbb{R}$, $\boldsymbol{f}_j \in \mathbb{R}^D$, and $\mathcal{U}_j \subset [N]$. In order to allow efficient optimization algorithms, we relax the $\{0, 1\}$-valued hard affiliation functions $\mathbb{1}_{\mathcal{U}}$ to functions that can assign soft affiliation values in $\mathbb{R}$. Letting $\chi$ denote the set of all indicator functions of the form $\mathbb{1}_{\mathcal{U}}$, for subsets $\mathcal{U} \in [N]$, we can formally define soft affiliation models as follows.

**Definition 3.1.** *A set $\mathcal{Q}$ of functions $\boldsymbol{q} : [N] \to \mathbb{R}$ that contains $\chi$ is called a* soft affiliation model.

**Definition 3.2.** *Let $d \in \mathbb{N}$, and let $\mathcal{Q}$ be a soft affiliation model. We define $[\mathcal{Q}] \subset \mathbb{R}^{N \times N} \times \mathbb{R}^{N \times D}$ to be the set of all elements of the form $(r\boldsymbol{q}\boldsymbol{q}^{\top}, \boldsymbol{q}\boldsymbol{f}^{\top})$, with $\boldsymbol{q} \in \mathcal{Q}$, $r \in \mathbb{R}$ and $\boldsymbol{f} \in \mathbb{R}^D$. We call $[\mathcal{Q}]$ the* soft rank-1 intersecting community graph (ICG) model *corresponding to $\mathcal{Q}$. Given $K \in \mathbb{N}$, the subset $[\mathcal{Q}]_K$ of $\mathbb{R}^{N \times N} \times \mathbb{R}^{N \times D}$ of all linear combinations of $K$ elements of $[\mathcal{Q}]$ is called the* soft rank-$K$ ICG model *corresponding to $\mathcal{Q}$.*

ICG models can be written in matrix form as follows. Given $K, D \in \mathbb{N}$, any intersecting community graph-signal $(\boldsymbol{C}, \boldsymbol{P}) \in \mathbb{R}^{N \times N} \times \mathbb{R}^{N \times D}$ in $[\mathcal{Q}]_K$ can be represented by a triplet of *community affiliation matrix* $\boldsymbol{Q} \in \mathbb{R}^{N \times K}$, *community magnitude vector* $\boldsymbol{r} \in \mathbb{R}^K$, and *community feature matrix* $\mathbf{F} \in \mathbb{R}^{K \times D}$. In the matrix form, $(\boldsymbol{C}, \boldsymbol{P}) \in [\mathcal{Q}]_K$ if and only if it has the form

$$\boldsymbol{C} = \boldsymbol{Q} \operatorname{diag}(\boldsymbol{r}) \boldsymbol{Q}^{\top} \quad \text{and} \quad \boldsymbol{P} = \boldsymbol{Q} \boldsymbol{F},$$

where $\operatorname{diag}(\boldsymbol{r})$ is the diagonal matrix in $\mathbb{R}^{K \times K}$ with $\boldsymbol{r}$ as its diagonal elements.

## 3.2 The Semi-Constructive Graph-Signal Weak Regularity Lemma

The following theorem is a semi-constructive version of the Weak Regularity Lemma for intersecting communities. It is semi-constructive in the sense that the approximating graphon is not just assumed to exist, but is given as the result of an optimization problem with an "easy to work with" loss, namely, the Frobenius norm error. The theorem also extends the standard weak regularity lemma by allowing soft affiliations to communities instead of hard affiliations. For comparison to previously poposed constructive regularity lemmas see Section 7.

**Theorem 3.1.** *Let $(\boldsymbol{A}, \boldsymbol{S})$ be a $D$-channel graph-signal of $N$ nodes, where $\deg(\boldsymbol{A}) = E'$. Let $K \in \mathbb{N}$, $\delta > 0$, and $\mathcal{Q}$ be a soft affiliation model. Consider the matrix-signal cut norm with weights $\alpha, \beta \geq 0$ not both zero, and the matrix-signal Frobenius norm with weights $\|(\boldsymbol{B}, \boldsymbol{X})\|_{\mathrm{F}} :=$ $\sqrt{\alpha \frac{N^2}{E} \|\boldsymbol{B}\|_{\mathrm{F}}^2 + \beta \|\boldsymbol{X}\|_{\mathrm{F}}^2}$. Let $m$ be sampled uniformly from $[K]$, and let $R \geq 1$ such that $K/R \in \mathbb{N}$.*

*Then, in probability $1 - \frac{1}{R}$ (with respect to the choice of $m$), for every $(\boldsymbol{C}^*, \boldsymbol{P}^*) \in [\mathcal{Q}]_m$,*

$$\text{if} \quad \|(\boldsymbol{A}, \boldsymbol{S}) - (\boldsymbol{C}^*, \boldsymbol{P}^*)\|_F \leq (1 + \delta) \min_{(\boldsymbol{C}, \boldsymbol{P}) \in [\mathcal{Q}]_m} \|(\boldsymbol{A}, \boldsymbol{S}) - (\boldsymbol{C}, \boldsymbol{P})\|_{\mathrm{F}}$$

$$\text{then} \quad \|(\boldsymbol{A}, \boldsymbol{S}) - (\boldsymbol{C}^*, \boldsymbol{P}^*)\|_{\square; N, E'} \leq \frac{3N}{2\sqrt{E'}} \sqrt{\frac{R}{K}} + \delta. \tag{6}$$

The proof of Theorem 3.1 is given in Appendix B, where the theorem is extended to graphon-signals. Theorem 3.1 means that we need not consider a complicated algorithm for estimating cut distance. Instead, we can optimize the Frobenius norm, which is much more direct (see Section 4). The term $\delta$ describes the stability of the approximation, where a perturbation in $(\boldsymbol{C}^*, \boldsymbol{P}^*)$ that corresponds to a small relative change in the optimal Frobenius error, leads to a small additive error in cut metric.

### 3.3 Approximation capabilities of ICGs

**ICGs vs. Message Passing.** Until the rest of the paper we suppose for simplicity that the degree of the graph $E'$ is equal to the number of edges $E$ up to a constant scaling factor $0 < \gamma \leq 1$, namely $E' = \gamma E$. Note that this is always true for unweighted graphs, where $\gamma = 1$. The bound in Theorem 3.1 is closer to be uniform with respect to $N$ the denser the graph is (the closer $E$ is to $N^2$). Denote the average node degree by $\mathrm{d} = E/N$. To get a meaningful bound in (6), we must choose $K > N^2/E$. On the other hand, for signal processing on the ICG to be more efficient than message passing, we require $K < \mathrm{d}$. Combining these two bounds, we see that ICG signal processing is guaranteed to be more efficient than message passing for any graph in the *semi-dense regime*: $\mathrm{d}^2 > N$ (or equivalently $E > N^{3/2}$).

**Essential tightness of Theorem 3.1.** In [2, Theorem 1.2] it was shown that the bound (6) is essentially tight in the following sense[4]. There exists a universal constant $c$ (greater than $1/2312$) such that for any $N$ there exists an unweighted graph $\boldsymbol{A} \in \mathbb{R}^{N \times N}$ with $\deg(\boldsymbol{A}) = E$, such that any $\boldsymbol{B} \in \mathbb{R}^{N \times N}$ that approximates $\boldsymbol{A}$ with error $\|\boldsymbol{A} - \boldsymbol{B}\|_{\square;N,E} \leq 16$ must have rank greater than $c\frac{N^2}{E}$. In words, there are graphs of $E$ edges that we cannot approximate in cut metric by any ICG with $K \ll \frac{N^2}{E}$. Hence, the requirement $K \gg N^2/E'$ for a small cut metric error in (6) for *any* graph is tight.

**ICGs approximations of sparse graphs.** In practice, many natural sparse graphs are amenable to low rank intersecting community approximations. For example, a simplistic model for how social networks are formed states that people connect according to shared characteristics (i.e. intersecting communities), like hobbies, occupation, age, etc [61]. Moreover, since ICGs can have negative community magnitudes $r_k$, one can also construct heterophilic components (bipartite substructures) of graphs with ICGs[5]. Hence, a social network can be naturally described using intersecting communities, even with $K < N^2/E$. For such graphs, ICG approximations are more accurate than their theoretical bound (6). In addition, Figure 5 in Appendix F.5 shows that in practice the Frobenius local minimizer, which does not give a small Frobenius error, guarantees small cut metric error even if $K < N^2/E$. This means that Frobenius minimization is a practical approach for fitting ICGs also for sparse natural graphs. Moreover, note that in the experiments in Tables 2 and 1 we take $K < N^2/E$ and still get competitive performance with SOTA message passing methods.

## 4 Fitting ICGs to graphs

Let us fix the soft affiliation model to be all vectors $\boldsymbol{q} \in [0,1]^N$. In this subsection, we propose algorithms for fitting ICGs to graphs based on Theorem 3.1.

### 4.1 Fitting an ICG to a graph with gradient descent

In view of Theorem 3.1, to fit a soft rank-$K$ ICG to a graph in cut metric, we learn a triplet $\boldsymbol{Q} \in [0,1]^{N \times K}$, $\boldsymbol{r} \in \mathbb{R}^K$ and $\boldsymbol{F} \in \mathbb{R}^{K \times D}$ that minimize the Frobenius loss:

$$L(\boldsymbol{Q}, \boldsymbol{r}, \boldsymbol{F}) = \left\| \boldsymbol{A} - \boldsymbol{Q} \operatorname{diag}(\boldsymbol{r}) \boldsymbol{Q}^\top \right\|_{\mathrm{F}}^2 + \lambda \left\| \boldsymbol{S} - \boldsymbol{Q} \boldsymbol{F} \right\|_{\mathrm{F}}^2, \tag{7}$$

where $\lambda \in \mathbb{R}$ is a scalar that balances the two loss terms. To implement $\boldsymbol{Q} \in [0,1]^{N \times K}$ in practice, we define $\boldsymbol{Q} = \operatorname{Sigmoid}(\boldsymbol{R})$ for a learned matrix $\boldsymbol{R} \in \mathbb{R}^{N \times K}$.

Suppose that $\boldsymbol{A}$ is sparse with $K^2, K\mathrm{d} \ll N$. The loss (7) involves sparse matrices and low rank matrices. Typical time complexity of sparse matrix operations is $\mathcal{O}(E)$, and for rank-$K$ matrices it

---

[4]We convert the result of [2] to our notations and definitions.

[5]Two sets of nodes (parts) $\mathcal{U}, \mathcal{V} \in [N]$ with connections between the parts but not within the parts can be represented by the ICG $\left( \mathbb{1}_{\mathcal{U} \cup \mathcal{V}} \mathbb{1}_{\mathcal{U} \cup \mathcal{V}}^\top - \mathbb{1}_{\mathcal{U}} \mathbb{1}_{\mathcal{U}}^\top - \mathbb{1}_{\mathcal{V}} \mathbb{1}_{\mathcal{V}}^\top \right)$.

is $\mathcal{O}(NK^2)$. Hence, we would like to derive an optimization procedure that takes $\mathcal{O}(K^2 N + E)$ operations. However, the first term of (7) involves a subtraction of the low rank matrix $\boldsymbol{Q}\operatorname{diag}(\boldsymbol{r})\boldsymbol{Q}^\top$ from the sparse matrix $\boldsymbol{A}$, which gives a matrix that is neither sparse nor low rank. Hence, a naïve optimization procedure would take $\mathcal{O}(N^2)$ operations. The next proposition shows that we can write the loss (7) in a form that leads to a $\mathcal{O}(K^2 N + KE)$ time and $\mathcal{O}(KN + E)$ space complexities.

**Proposition 4.1.** *Let $\boldsymbol{A} = (a_{i,j})_{i,j=1}^N$ be an adjacency matrix of a weighted graph with $E$ edges. The graph part of the Frobenius loss can be written as*

$$\left\| \boldsymbol{A} - \boldsymbol{Q}\operatorname{diag}(\boldsymbol{r})\boldsymbol{Q}^\top \right\|_{\mathrm{F}}^2 = \frac{1}{N^2} \operatorname{Tr}\left( (\boldsymbol{Q}^\top \boldsymbol{Q})\operatorname{diag}(\boldsymbol{r})(\boldsymbol{Q}^\top \boldsymbol{Q})\operatorname{diag}(\boldsymbol{r}) \right) + \|\boldsymbol{A}\|_{\mathrm{F}}^2$$

$$- \frac{2}{N^2} \sum_{i=1}^N \sum_{j \in \mathcal{N}(i)} \boldsymbol{Q}_{i,:}\operatorname{diag}(\boldsymbol{r})\left(\boldsymbol{Q}^\top\right)_{:,j} a_{i,j}.$$

*Computing the right-hand-side and its gradients with respect to $\boldsymbol{Q}$ and $\boldsymbol{r}$ has a time complexity of $\mathcal{O}(K^2 N + KE)$, and a space complexity of $\mathcal{O}(KN + E)$.*

The proof is in Appendix C. We optimize $\boldsymbol{Q}, \boldsymbol{r}$, and $\boldsymbol{F}$ using gradient descent (GD) on (7) implemented via Proposition 4.1. After convergence, we further refine $\boldsymbol{F}$ by setting $\boldsymbol{F} = \boldsymbol{Q}^\dagger \boldsymbol{S}$.

## 4.2  Initialization

In Appendix D we propose a good initialization for the GD minimization of (7). Suppose that $K$ is divisible by 3. Let $\boldsymbol{\Phi}_{K/3} \in \mathbb{R}^{N \times K/3}$ be the matrix consisting of the leading eigenvectors of $\boldsymbol{A}$ as columns, and $\boldsymbol{\Lambda}_{K/3} \in \mathbb{R}^{K/3 \times K/3}$ the diagonal matrix of the leading eigenvalues. For each eigenvector $\phi$ with eigenvalue $\lambda$, consider the three soft indicators

$$\phi_1 = \frac{\phi_+}{\|\phi_+\|_\infty}, \quad \phi_2 = \frac{\phi_-}{\|\phi_-\|_\infty}, \quad \phi_3 = \frac{\phi_+ + \phi_-}{\|\phi_+ + \phi_-\|_\infty} \tag{8}$$

with community affiliation magnitudes $r_1 = 2\lambda\|\phi_+\|_\infty^2$, $r_2 = 2\lambda\|\phi_-\|_\infty^2$ and $r_3 = -\lambda\|\phi_+ + \phi_-\|_\infty^2$ respectively. Here, $\phi_\pm$ is the positive or negative part of the vector. One can now show that the ICG $\boldsymbol{C}$ based on the $K$ soft affiliations corresponding to the leading $K/3$ eigenvectors approximates $\boldsymbol{A}$ in cut metric with error $\mathcal{O}(K^{-1/2})$. Note that computing the leading eigenvectors is efficient with any variant of the power iteration for sparse matrices, like any variant of Lanczos algorithm, which takes $\mathcal{O}(E)$ operations per iteration, and requires just a few iteration due to its fast super-exponential convergence [47]. We moreover initialize $\boldsymbol{F}$ optimally by $\boldsymbol{F} = \boldsymbol{Q}^\dagger \boldsymbol{S}$.

## 4.3  Subgraph SGD for fitting ICGs to large graphs

In many situations, we need to process large graphs for which $E$ is too high to read to the GPU memory, but $NK$ is not. These are the situation where processing graph-signals using ICGs is beneficial. However, to obtain the ICG we first need to optimize the loss (7) using a message-passing type algorithm that requires reading $E$ edges to memory. To allow such processing, we next show that one can read the $E$ edges to shared RAM (or storage), and at each SGD step read to the GPU memory only a random subgraph with $M$ nodes.

At each interation, we sample $M \ll N$ random nodes $\boldsymbol{n} := (n_m)_{m=1}^M$ uniformly and independently from $[N]$ (with repetition). We construct the sub-graph $\boldsymbol{A}^{(\boldsymbol{n})} \in \mathbb{R}^{M \times M}$ with entries $a_{i,j}^{(\boldsymbol{n})} = a_{n_i,n_j}$, and the sub-signal $\boldsymbol{S}^{(\boldsymbol{n})} \in \mathbb{R}^{M \times K}$ with entries $\boldsymbol{s}_i^{(\boldsymbol{n})} = \boldsymbol{s}_{n_i}$. We similarly define the sub-community affiliation matrix $\boldsymbol{Q}^{(\boldsymbol{n})} \in [0,1]^{M \times K}$ with entries $q_{i,j}^{(\boldsymbol{n})} = q_{n_i,j}$. We consider the loss

$$L^{(\boldsymbol{n})}(\boldsymbol{Q}^{(\boldsymbol{n})}, \boldsymbol{r}, \boldsymbol{F}) = \left\| \boldsymbol{A}^{(\boldsymbol{n})} - \boldsymbol{Q}^{(\boldsymbol{n})}\operatorname{diag}(\boldsymbol{r})\boldsymbol{Q}^{(\boldsymbol{n})\top} \right\|_{\mathrm{F}}^2 + \lambda \left\| \boldsymbol{S}^{(\boldsymbol{n})} - \boldsymbol{Q}^{(\boldsymbol{n})}\operatorname{diag}(\boldsymbol{r})\boldsymbol{F} \right\|_{\mathrm{F}}^2 \tag{9}$$

which depends on all entries of $\boldsymbol{F}$ and $\boldsymbol{r}$ and on the $\boldsymbol{n}$ entries of $\boldsymbol{Q}$. Each SGD updates all of the entries of $\boldsymbol{F}$ and $\boldsymbol{r}$, and the $\boldsymbol{n}$ entries of $\boldsymbol{Q}$ by incrementing them with the respective gradients of $L^{(\boldsymbol{n})}(\boldsymbol{Q}^{\boldsymbol{n}}, \boldsymbol{r}, \boldsymbol{F})$. Hence, $\nabla_{\boldsymbol{Q}} L^{(\boldsymbol{n})}(\boldsymbol{Q}^{(\boldsymbol{n})}, \boldsymbol{r}, \boldsymbol{F})$ may only be nonzero for entries $\boldsymbol{q}_n$ with $n \in \boldsymbol{n}$. Proposition E.1 in Appendix E shows that the gradients of $L^{(\boldsymbol{n})}$ approximate the gradients of $L$. More concretely, $\frac{M}{N}\nabla_{\boldsymbol{Q}^{(\boldsymbol{n})}} L^{(\boldsymbol{n})} \approx \nabla_{\boldsymbol{Q}^{(\boldsymbol{n})}} L$, $\nabla_{\boldsymbol{r}} L^{(\boldsymbol{n})} \approx \nabla_{\boldsymbol{r}} L$, and $\nabla_{\boldsymbol{F}} L^{(\boldsymbol{n})} \approx \nabla_{\boldsymbol{F}} L$. Note that the stochastic gradients with respect to $\boldsymbol{Q}^{(\boldsymbol{n})}$ approximate a scaled version of the full gradients.

# 5 Learning with ICG

Given a graph-signal $(\boldsymbol{A}, \boldsymbol{S})$ and its approximating ICG $(\boldsymbol{C}, \boldsymbol{P})$, the corresponding soft affiliations $\boldsymbol{Q} = (\boldsymbol{q}_k)_{k=1}^K$ represent intersecting communities that can describe the adjacency structure $\boldsymbol{A}$ and can account for the variability of $\boldsymbol{S}$. In a deep network architecture, one computes many latent signals, and these signals typically correspond in some sense to the structure of the data $(\boldsymbol{A}, \boldsymbol{S})$. It is hence reasonable to put a special emphasis on latent signals that can be described as linear combinations of $(\boldsymbol{q}_k)_{k=1}^K$. Next, we develop a signal processing framework for such signals.

**Graph signal processing with ICG.** We develop a signal processing approach with $\mathcal{O}(NK)$ complexity for each elemental operation. Let $\boldsymbol{Q} \in \mathbb{R}^{N \times K}$ be a fixed community matrix. We call $\mathbb{R}^{N \times D}$ the *node space* and $\mathbb{R}^{K \times D}$ the *community space*. We process signals via the following operations:

- *Synthesis* is the mapping $\boldsymbol{F} \mapsto \boldsymbol{QF}$ from the community space to the node space, in $\mathcal{O}(NKD)$.
- *Analysis* is the mapping $\boldsymbol{S} \mapsto \boldsymbol{Q}^\dagger \boldsymbol{S}$ from the node space to the community space, in $\mathcal{O}(NKD)$.
- *Community processing* refers to any operation that manipulates the community feature vector $\boldsymbol{F}$ (e.g., an MLP in which a linear operator requires $KD \times KD$ parameters or a Transformer) in $\mathcal{O}(K^2 D^2)$ operations.
- *Node processing* is any function $\Theta$ that operates in the node space on nodes independently via $\Theta(\boldsymbol{S}) := (\theta(\boldsymbol{s}_n))_{n=1}^N$, where $\theta : \mathbb{R}^D \to \mathbb{R}^{D'}$ (e.g., an MLP) takes $\mathcal{O}(DD')$ operations. Node processing has time complexity of $\mathcal{O}(NDD')$.

Analysis and synthesis satisfy the reconstruction formula $\boldsymbol{Q}^\dagger \boldsymbol{QF} = \boldsymbol{F}$. Moreover, $\boldsymbol{QQ}^\dagger \boldsymbol{S}$ is the projection of $\boldsymbol{S}$ upon the space of linear combinations of communities $\{\boldsymbol{QF} \mid \boldsymbol{F} \in \mathbb{R}^{K \times D}\}$. Note that when $\boldsymbol{A}$ is not low rank, since $\boldsymbol{Q}$ is almost surely full rank when initialized randomly, the optimal configuration of $\boldsymbol{Q}$ would avoid having repetitions of communities, as this would unnecessarily reduce its rank. Therefore, $\boldsymbol{Q}$ is typically full rank, and in this generic case $\boldsymbol{Q}^\dagger = (\boldsymbol{Q}^\top \boldsymbol{Q})^{-1} \boldsymbol{Q}^\top$.

**Deep ICG Neural Networks.** In the following, we propose deep architectures based on the elements of signal processing with an ICG, which take $\mathcal{O}(D(NK + K^2 D + ND))$ operations at each layer. In comparison, simplified message passing layers (e.g., GCN and GIN), where the message is computed using just the feature of the transmitting node, have a complexity of $\mathcal{O}(ED + ND^2)$. General message passing layers, which define messages via a function applied on the concatenated pair of node features of each edge, take $\mathcal{O}(ED^2)$ operations for MLP message functions. Therefore, ICG neural networks are more efficient than MPNNs if $K \ll D\mathrm{d}$, where d is the average node degree.

We denote by $D^{(\ell)}$ the dimension of the node features at layer $\ell$ and define the initial node representations $\boldsymbol{H}^{(0)} = \boldsymbol{S}$ and the initial community features $\boldsymbol{F}^{(0)} = \boldsymbol{Q}^\dagger \boldsymbol{S}$. The node features at layers $0 \le \ell \le L - 1$ are defined by

$$\boldsymbol{H}^{(\ell+1)} = \sigma \left( \boldsymbol{H}^{(\ell)} \boldsymbol{W}_1^{(\ell)} + \boldsymbol{Q} \Theta \left( \boldsymbol{F}^{(\ell)} \right) \boldsymbol{W}_2^{(\ell)} \right), \qquad \boldsymbol{F}^{(\ell+1)} = \boldsymbol{Q}^\dagger \boldsymbol{H}^{(\ell)},$$

where $\Theta$ is a neural network (i.e. multilayer perceptron or MultiHeadAttention), $\boldsymbol{W}_1^{(\ell)}, \boldsymbol{W}_2^{(\ell)} \in \mathbb{R}^{D^{(\ell)} \times D^{(\ell+1)}}$ are learnable matrices acting on signals in the node space, and $\sigma$ is a non-linearity. We call this method ICG neural network (ICG-NN). The final representations $\boldsymbol{H}^{(L)}$ can be used for predicting node-level properties.

We propose another deep ICG method, where $\boldsymbol{F}^{(\ell)} \in \mathbb{R}^{K \times D^{(\ell)}}$ are taken directly as trainable parameters. Namely,

$$\boldsymbol{H}^{(\ell+1)} = \sigma \left( \boldsymbol{H}^{(\ell)} \boldsymbol{W}_s^{(\ell)} + \boldsymbol{Q} \boldsymbol{F}^{(\ell)} \boldsymbol{W}^{(\ell)} \right).$$

We call this method $\text{ICG}_{\text{u}}$-NN for the *unconstrained* community features. The motivation is that $\boldsymbol{Q} \boldsymbol{F}^{(\ell)}$ exhausts the space of all signals that are linear combinations of the communities. If the number of communities is not high, then this is a low dimensional space that does not lead to overfitting in typical machine learning problems. Therefore, there is no need to reduce the complexity of $\boldsymbol{F}^{(\ell)}$ by constraining it to be the output of a neural network $\Theta$ on latent representations. The term $\boldsymbol{Q} \boldsymbol{F}^{(\ell)}$ can hence be interpreted as a type of positional encoding that captures the community structure.

**ICG-NNs for spatio-temporal graphs.** For a fixed graphs with varying node features, the ICG is fitted to the graph once[6]. Given that there are $T$ training signals, each learning step with ICG-NN takes

---

[6]We either ignore the signal in the loss, or concatenate random training signals and reduce their dimension to obtain one signal with low dimension $D$. This signal is used as the target for the ICG.

$\mathcal{O}(TN(K + D))$ rather than $\mathcal{O}(TED^2)$ for MPNNs. Note that $T$ does not scale the preprocessing time. Hence, the time dimension amplifies the difference in efficiency between ICG-NNs and MPNNs.

## 6 Experiments

We empirically validate our methods with the following experiments: [7]

- **Runtime analysis** (Section 6.1): We report the forward pass runtimes of $ICG_u$-NN and GCN [30], empirically validating the theoretical advantage of the former. We further extend this analysis in Appendices F.7 and F.8.
- **Node classification** (Appendix F.1): We evaluate our method on real-world node classification datasets [43, 45, 36], observing that the model performance is competitive with standard approaches.
- **Node classification using Subgraph SGD** (Section 6.2 and Appendix F.3): We evaluate our subgraph SGD method (Section 4.3) to identify the effect of sampling on the model performance on the tolokers and Flickr datasets [43, 65]. We find the model's performance to be robust on tolokers and state-of-the-art on Flickr.
- **Spatio-temporal tasks** (Section 6.3): We evaluate $ICG_u$-NN on real-world spatio-temporal tasks [35] and obtain competitive performance to domain-specific baselines.
- **Comparison to graph coarsening methods** (Appendix F.2): We provide an empirical comparison between ICG-NNs and a variety of graph coarsening methods on the Reddit [23] and Flickr [65] datasets, where ICG-NNs achieve state-of-the-art performance.
- **Additional experiments:** We perform an ablation study over the number of communities (Appendix F.4) and the choice of initialization in Section 4.2 (Appendix F.6). We moreover experimentally demonstrate a positive correlation between the Frobenius error and cut norm error as hinted by Theorem 3.1 (Appendix F.5), and perform a memory allocation analysis (Appendix F.9).

### 6.1 How does the runtime compare to standard GNNs?

**Setup.** We compare the forward pass runtimes of our signal processing pipeline ($ICG_u$-NN) and GCN [30] on *Erdős-Rényi* $ER(n, p(n) = 0.5)$ graphs with up to $7k$ nodes. Node features are independently drawn from $U[0, 1]$ and the initial feature dimension is $128$. Both models use a hidden dimension of $128$, $3$ layers and an output dimension of $5$.

**Results.** Figure 2 reveals a strong square root relationship between the runtime of $ICG_u$-NN and the runtime of GCN. This aligns with our expectations, as the time complexity of GCN is $\mathcal{O}(E)$, while the time complexity of ICG-NNs is $\mathcal{O}(N)$, highlighting the computational advantage of using ICGs. We complement this analysis with experiments using sparse *Erdős-Rényi* graphs in Appendix F.7.

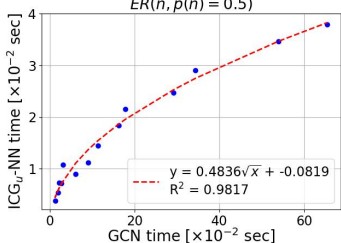

Figure 2: Runtime of K-$ICG_u$-NN (for K=100) as a function of GCN forward pass duration on graphs $G \sim ER(n, p(n) = 0.5)$.

### 6.2 Node classification using Subgraph SGD

**Setup.** We evaluate $ICG_u$-NN on the non-sparse graphs tolokers [43], following the 10 data splits of [43]. We report the mean ROC AUC and standard deviation as a function of the ratio of nodes that are removed from the graph. We compare to the baseline of MLP on the full graph.

**Results.** Figure 3 shows a slight degradation of 2.8% when a small number of nodes is removed from the graph. However, the key insight is that when more than 10% of the graph is removed, the performance stops degrading. These results further support our Proposition E.1 in Appendix E about the error between subgraph gradients and full-graph gradients of the Frobenius loss, and establish ICG-NNs as a viable option for learning on large graphs.

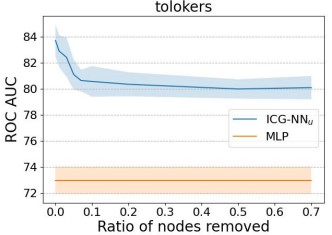

Figure 3: ROC AUC of $ICG_u$-NN and an MLP as a function of the % nodes removed from the graph.

---

[7]All our experiments are run on a single NVidia L40 GPU. We made our codebase available online: https://github.com/benfinkelshtein/ICGNN.

## 6.3 Spatio-temporal graphs

**Setup & baselines.** We evaluate ICG-NN and ICG$_\text{u}$-NN on real-world traffic networks, METR-LA and PEMS-BAY [35] following the methodology described by [15]. We segment the datasets into windows of 12 time steps and train the models to predict the subsequent 12 observations. For all datasets, these windows are divided sequentially into 70% for training, 10% for validation, and 20% for testing. We report the mean absolute error (MAE) and standard deviation averaged over the forecastings. The baselines DCRNN [35], GraphWaveNet [60], AGCRN [8], T&S-IMP, TTS-IMP, T&S-AMP, and TTS-AMP [15], are adopted from [15]. We incorporate a GRU to embed the data before inputting it into our ICG-NN models, we symmetrize the graph to enable our method to operate on it, and we disregard the features in the ICG approximation (setting $\lambda = 0$ in (7)). Additionally, we use the Adam optimizer and detail all hyperparameters in Appendix I.

Table 1: Results on dense temporal graphs. Top three models are colored by First, Second, Third.

| relative Frob. | METR-LA 0.44 | PEMS-BAY 0.34 |
|---|---|---|
| DCRNN | $3.22 \pm 0.01$ | $1.64 \pm 0.00$ |
| GraphWaveNet | $3.05 \pm 0.03$ | $1.56 \pm 0.01$ |
| AGCRN | $3.16 \pm 0.01$ | $1.61 \pm 0.00$ |
| T&S-IMP | $3.35 \pm 0.01$ | $1.70 \pm 0.01$ |
| TTS-IMP | $3.34 \pm 0.01$ | $1.72 \pm 0.00$ |
| T&S-AMP | $3.22 \pm 0.02$ | $1.65 \pm 0.00$ |
| TTS-AMP | $3.24 \pm 0.01$ | $1.66 \pm 0.00$ |
| ICG-NN | $3.42 \pm 0.03$ | $1.76 \pm 0.00$ |
| ICG$_\text{u}$-NN | $3.12 \pm 0.01$ | $1.56 \pm 0.00$ |

**Results.** Table 1 shows that ICG-NNs achieve competitive performance when compared to methods that are specially tailored for spatio-temporal data such as DCRNN, GraphWaveNet and AGCRN. Despite the small graph size (207 and 325 nodes) and the low ratio of edges (graph densities of $3.54 \cdot 10^{-2}$ and $2.24 \cdot 10^{-2}$), ICG-NNs perform well, corroborating the discussion in Section 3.3.

## 7 Related work

We provide the main related work in this Section. In Appendix G we give an additional review.

**Intersecting communities and stochastic block models.** We express graphs as intersections of cliques, or communities, similarly to classical works in statistics and computer science [1, 62]. Our approach can be interpreted as fitting a stochastic-block-model (SBM) to a graph. As opposed to standard SBM approaches, our method is interpreted as data fitting with norm minimization rather than statistical inference. Similarly to the intersecting community approach of BigCLAM [62], our algorithm takes $\mathcal{O}(E)$ operations. Unlike BigCLAM, however, we can approximate any graph, as guaranteed by the regularity lemma. This is possible since ICGs are allowed to have negative coefficients, while BigCLAM only uses positive coefficients due to its probabilistic interpretation. To the best of our knowledge, we are the first to propose a SBM fitting algorithm based on the weak regularity lemma. For a survey on SBMs we refer the reader to [32].

**The Weak Regularity Lemma.** The Regularity Lemma is a central result in graph theory with many variants and many proof techniques. One version is called the *Weak Regularity Lemma* (for graphs [20], graphons [38], or graph-signals and graphon-signals [33]), and has the following interpretation: every graph can be approximated in the cut metric by an ICG, where the error rate $\epsilon$ is uniformly $\mathcal{O}(K^{-1/2})$[8]. While [20, Theorem 2] proposes an algorithm for finding the approximating ICG, the algorithm takes $N2^{\mathcal{O}(K)}$ time to find this minimizer[9]. This time complexity is too high to be practical in real-world problems. Alternatively, since the cut metric is defined via a maximization process, finding a minimizing ICG by directly optimizing cut metric via a GD approach involves a min-max problem, which appears numerically problematic in practice. Moreover, computing cut norm is NP-hard [3, 44]. Instead, we consider a way to bypass the need to explicitly compute the cut norm, finding a $K$ community ICG with error $\mathcal{O}(K^{-1/2})$ in the cut metric by minimizing the Frobenius norm. Here, each gradient descent step takes $O(EK)$ operations, and $EK$ is typically much smaller than $N2^K$. As opposed to the algorithm in [20], our theorem is only semi-constructive, as GD is not guaranteed to find the global minimum of the Frobenius error. Still, our theorem motivates a *practical* strategy for estimating ICGs, while the algorithm formulated in [20] is only of theoretical significance. We note that for the Szemerédi (non-weak) regularity lemma [49], it was shown by Alon

---

[8]In some papers the lemma is formulated for non-intersecting blocks $\sum_{i,j} r_{i,j} \mathbb{1}_{\mathcal{U}_i} \mathbb{1}_{\mathcal{U}_j}$, where $\cup \{\mathcal{U}_j\} = [N]$, with error $\epsilon = \mathcal{O}\big((\log(K))^{-1/2}\big)$. The intersecting community case is an intermediate step of the proof.

[9]One needs to convert the scaling of the Frobenius and cut norms used in [20] to our scaling and to follow the proof of [20, Theorem 2] to see this.

et al. [4] that a regular partition with error $< \epsilon$ into non-intersecting communities (the analogue to ICG in the Szemerédi regularity lemma) can be found in polynomial time with respect to $N$, provided that $N$ is very large ($N \sim 2^{\epsilon^{-20}}$). For more details on the Weak Regularity Lemma, see Appendix A.

**Subgraph methods.** Learning on large graphs requires sophisticated subgraph sampling techniques for deployment on contemporary processors [24, 12, 13, 7, 65]. After the preprocessing step on the ICG, our approach allows processing very large networks more accurately, avoiding subsampling schemes that can alter the properties of the graph in unintended ways.

**GNNs with local pooling.** Local graph pooling methods, e.g., [63, 9], construct a sequence of coarsened versions of the given graph, each time collapsing small sets of neighboring nodes into a single "super-node." At each local pooling layer, the signal is projected from the finer graph to the coarser graph. This is related to our ICG-NN approach, where the signal is projected upon the communities. As opposed to local graph pooling methods, our communities intersect and have large-scale supports. Moreover, our method does not lose the granular/high-frequency content of the signal when projecting upon the communities, as we also have node-wise operations at the fine level. In pooling approaches, the graph is partitioned into disjoint, or slightly overlapping communities, each community collapses to a node, and a standard message passing scheme is applied on this coarse graph. In ICG-NNs, each operation on the community features has a global receptive field in the graph. Moreover, ICG-NNs are not of message-passing type: the (flattened) community feature vector $\boldsymbol{F}$ is symmetryless and is operated upon by a general MLP in the ICG-NN, while MPNNs operate by the same function on all edges.

In terms of computational efficiency, GNNs with local pooling do not asymptotically improve runtime, as the first layer operates on the full graph, while our method operates solely on the efficient data structure.

**Variational graph autoencoders.** Our ICG construction is related to statistical network analysis with latent position graph models [6]. Indeed, the edge weight between any pair of nodes $n, m$ in the ICG is the diagonal Bilinear product between their affiliation vectors, namely, $\sum_j r_j \mathbb{1}_{\mathcal{U}_j}(n) \mathbb{1}_{\mathcal{U}_j}(m)$, which is similar to the inner produce decoder in variational graph autoencoders [29, 6]. However, as opopsed to variational graph autoencoders, we have the coefficients $r_j$, that can be negative and hence allow a more expressive decoding.

# 8    Summary

We introduced an new approach for learning on large non-sparse graphs, using ICGs. We proved a new constructive variant of the weak regularity lemma, which shows that minimizing the Frobenius error between the ICG and the graph leads to a uniformly small error in cut metric. We moreover showed how to optimize the Frobenius error efficiently. We then developed a signal processing setting, operating on the ICG and on the node reprsentations in $\mathcal{O}(N)$ complexity. The overall pipeline involves precomputing the ICG approximation of the graph in $\mathcal{O}(E)$ operations per iteration, and then solving the task on the graph in $\mathcal{O}(N)$ operations per iteration. Both fitting an ICG to a large graph, and training a standard subgraph GNNs, require an online subsampling method, reading from slow memory during optimization. However, fitting the ICG is only done once, and does not require an extensive hyperparameter optimization. Then, learning to solve the task on the graph with ICG-NNs is efficient, and can be done directly on the GPU memory. Since the second learning phase is the most demanding part, involving an extensive hyperparameter optimization and architecture tuning, ICG-NN offer a potentially significant advantage over standard subgraph GNNs. This gap between ICG-NNs and MPNNs is further amplified for time series on graphs.

The main limitation of our method is that the ICG-NN is fitted to a specific ICG, and cannot be naïvely transferred between different ICGs approximating different graphs. Another limitation is the fact that the current ICG construction is limited to undirected graphs, while many graphs, especially spatiotemporal, are directed. One potential avenue for future work is thus to extend the ICG construction to directed graphs. Additional future work will study the expressive power of ICGs.

**Acknowledgement**
The first author is funded by the Clarendon scholarship. This research was supported by the Israel Science Foundation (grant No. 1937/23) and the United States - Israel Binational Science Foundation (NSF-BSF, grant No. 2024660). This work was also partially funded by EPSRC Turing AI World-Leading Research Fellowship No. EP/X040062/1.

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

# A The weak regularity lemma

Let $G = \{\mathcal{N}, \mathcal{E}\}$ be a graph and $\mathcal{P} = \{\mathcal{N}_1, \ldots, \mathcal{N}_k\}$ be a partition of $\mathcal{N}$. The partition is called *equipartition* if $||\mathcal{N}_i| - |\mathcal{N}_j|| \leq 1$ for every $1 \leq i, j \leq k$. For any two subsets $\mathcal{U}, \mathcal{S} \subset \mathcal{N}$, the number of edges from $\mathcal{U}$ to $\mathcal{S}$ are denoted by $e_G(\mathcal{U}, \mathcal{S})$. Given two node set $\mathcal{U}, \mathcal{S} \subset \mathcal{N}$, if the edges between $\mathcal{N}_i$ and $\mathcal{N}_j$ were distributed randomly, then, the number of edges between $\mathcal{U}$ and $\mathcal{S}$ would have been close to the expected value

$$e_{\mathcal{P}(\mathcal{U},\mathcal{S})} := \sum_{i=1}^{k} \sum_{j=1}^{k} \frac{e_G(\mathcal{N}_i, \mathcal{N}_j)}{|\mathcal{N}_i| |\mathcal{N}_j|} |\mathcal{N}_i \cap \mathcal{U}| |\mathcal{N}_j \cap \mathcal{S}|.$$

Thus, the *irregularity*, which measure of how non-random like the edges between $\{\mathcal{N}_j\}$ are, is defined to be

$$\mathrm{irreg}_G(\mathcal{P}) = \max_{\mathcal{U},\mathcal{S} \subset \mathcal{N}} |e_G(\mathcal{U}, \mathcal{S}) - e_{\mathcal{P}}(\mathcal{U}, \mathcal{S})| / |\mathcal{N}|^2. \tag{10}$$

**Theorem A.1** (Weak Regularity Lemma [20])**.** *For every $\epsilon > 0$ and every graph $G = (\mathcal{N}, \mathcal{E})$, there is an equipartition $\mathcal{P} = \{\mathcal{N}_1, \ldots, \mathcal{N}_k\}$ of $\mathcal{N}$ into $k \leq 2^{c/\epsilon^2}$ classes such that $\mathrm{irreg}_G(\mathcal{P}) \leq \epsilon$. Here, $c$ is a universal constant that does not depend on $G$ and $\epsilon$.*

The weak regularity lemma states that any large graph $G$ can be represented by the weighted graph $G^\epsilon$ with node set $\mathcal{N}^\epsilon = \{\mathcal{N}_1, \ldots, \mathcal{N}_k\}$, where the edge weight between the nodes $\mathcal{N}_i$ and $\mathcal{N}_j$ is $\frac{e_G(\mathcal{N}_i, \mathcal{N}_j)}{|\mathcal{N}_i|, |\mathcal{N}_j|}$, which depicts a smaller, coarse-grained version of the large graph. The "large-scale" structure of $G$ is given by $G^\epsilon$, and the number of edges between any two subsets of nodes $\mathcal{U}_i \subset \mathcal{N}_i$ and $\mathcal{U}_j \subset \mathcal{N}_j$ is close to the "expected value" $e_{\mathcal{P}(\mathcal{U}_i, \mathcal{U}_j)}$. Hence, the deterministic graph $G$ "behaves" as if it was randomly sampled from the "stochastic block model" $G^\epsilon$.

It can be shown that irregularity coincides with error in cut metric between the graph and its coarsening SBM. Namely, to see how the cut-norm is related to irregularity (10), consider a graphon $W_G$ induced by the graph $G = \{\mathcal{N}, \mathcal{E}\}$. Let $\mathcal{P} = \{\mathcal{N}_1, \ldots, \mathcal{N}_k\}$ be an equipartition of $\mathcal{N}$. Consider the graph $G^{\mathcal{P}}$ defined by the adjacency matrix $A^{\mathcal{P}} = (a_{i,j}^{\mathcal{P}})_{i,j=1}^{|\mathcal{N}|}$ with

$$a_{i,j}^{\mathcal{P}} = \frac{e_G(\mathcal{N}_{q_i}, \mathcal{N}_{q_j})}{|\mathcal{N}_{q_i}|, |\mathcal{N}_{q_j}|},$$

where $\mathcal{N}_{q_i} \in \mathcal{P}$ is the class that contains node $i$. Now, it can be verified that

$$\|W_G - W_{G^{\mathcal{P}}}\|_{\square} = \mathrm{irreg}_G(\mathcal{P}).$$

Hence, the weak regularity lemma can be formulated with cut norm instead of irregularity.

# B Graphon-Signal Intersecting Communities and the Constructive Regularity Lemma

In this section we prove a constructive weak regularity lemma for graphon-signals, where Theorem 3.1 is a special case. We start by defining graphon-signals in Subsection B.1. We define graphon-signal cut norm and metric in Subsection B.2, and graphon-signal intersecting communities in B.3. In Subsections B.4 and B.5 we formulate and prove the constructive graphon-signal weak regularity lemma. Finally, in Subsection B.6 we prove the constructive graph-signal weak regularity lemma as a special case.

For $m, J \in \mathbb{N}$, the $L_2$ norm of a measurable function $f = (f_i)_{i=1}^{j} : [0,1]^m \to \mathbb{R}^J$, where $f_i : [0,1]^m \to \mathbb{R}$, is defined to be $\|f\|_2 = \sqrt{\sum_{j=1}^{J} \int_{[0,1]^m} |f_j(\boldsymbol{x})|^2 \, d\boldsymbol{x}}$.

## B.1 Graphon-signals

A graphon [11, 37] can be seen as a weighted graph with a "continuous" node set $[0,1]$. The space of graphons $\mathcal{W}$ is defined to be the set of all measurable symmetric function $W : [0,1]^2 \to [0,1]$, $W(x,y) = W(y,x)$. Each *edge weight* $W(x,y) \in [0,1]$ of a graphon $W \in \mathcal{W}$ can be seen as the

probability of having an edge between nodes $x$ and $y$. Graphs can be seen as special graphons. Let $\mathcal{I}_m = \{I_1, \ldots, I_m\}$ be an *interval equipartition*: a partition of $[0,1]$ into disjoint intervals of equal length.

A graph with an adjacency matrix $\boldsymbol{A} = (a_{i,j})_{i,j=1}^N$ *induces* the graphon $W_{\boldsymbol{A}}$, defined by $W_{\boldsymbol{A}}(x,y) = a_{\lceil xm \rceil, \lceil ym \rceil}$ [10]. Note that $W_{\boldsymbol{A}}$ is piecewise constant on the partition $\mathcal{I}_m$. We hence identify graphs with their induced graphons. A graphon can also be seen as a generative model of graphs. Given a graphon $W$, a corresponding random graph is generated by sampling i.i.d. nodes $\{u_n\}$ from the graphon domain $[0,1]$, and connecting each pair $u_n, u_m$ in probability $W(u_n, u_m)$ to obtain the edges of the graph.

The space of grahon-signals is the space of pairs of measurable functions $(W, s)$ of the form $W : [0,1]^2 \to [0,1]$ and $s : [0,1] \to [0,1]^D$, where $D \in \mathbb{N}$ is the number of signal channels. Note that datasets with signals in value ranges other than $[0,1]$ can always be transformed (by translating and scaling in the value axis of each channel) to be $[0,1]^D$-valued. Given a discrete signal $\boldsymbol{S} : [N] \to [0,1]^D$, we define the induced signal $s_{\boldsymbol{S}}$ over $[0,1]$ by $s_{\boldsymbol{S}}(x) = s_{\lceil xm \rceil}$. We define the *Frobenius distance* between two graphon-signals $(W, s)$ and $(W', s')$ simply as their $L_2$ distance, namely,

$$\|(W, s) - (W', s')\|_{\mathrm{F}} := \sqrt{a \|W - W'\|_2^2 + b \sum_{j=1}^D \|s_j - s'_j\|_2^2} \tag{11}$$

for some $a, b > 0$.

## B.2 Cut-distance

The *cut metric* is a natural notion of graph similarity, based on the *cut norm*. The graphon-signal cut norm was defined in [33], extending the standard definition to a graphon with node features.

**Definition B.1.** *The graphon cut norm of $W \in L^2[0,1]^2$ is defined to be*

$$\|W\|_\square = \sup_{S,T \subset [0,1]} \left| \int_S \int_T W(x,y) dx dy \right|. \tag{12}$$

*The signal cut norm of $s = (s_1, \ldots, s_D) \in (L^2[0,1])^D$ is defined to be*

$$\|s\|_\square = \frac{1}{D} \sum_{j=1}^D \sup_{U \subset [0,1]} \left| \int_U s_j(x) dx \right|. \tag{13}$$

*The graphon-signal cut norm of $(W, s) \in L^2[0,1]^2 \times (L^2[0,1])^D$, with weights $\alpha > 0$ and $\beta > 0$, is defined to be*

$$\|(W, s)\|_\square = \alpha \|W\|_\square + \beta \|s\|_\square. \tag{14}$$

Given two graphons $W, W'$, their distance in *cut metric* is the *cut norm* of their difference, namely

$$\|W - W'\|_\square = \sup_{S,T \subset [0,1]} \left| \int_S \int_T \left( W(x,y) - W'(x,y) \right) dx dy \right|. \tag{15}$$

The right-hand-side of (15) is interpreted as the similarity between the adjacencies $W$ and $W'$ on the block on which they are the most dissimilar. The *graphon-signal cut metric* is similarly defined to be $\|(W, s) - (W', s')\|_\square$. The cut metric between two graph-signals is defined to be the cut metric between their induced graphon-signals.

In Subsection B.6 we show that the graph-signal cut norm of Definition 2.1 are a special case of graphon-signal cut norm for induced graph-signals.

## B.3 Intersecting community graphons

Here, we define ICGs for graphons. Denote by $\chi$ the set of all indicator function of measurable subset of $[0,1]$

$$\chi = \{\mathbb{1}_u \mid u \subset [0,1] \text{ measurable}\}.$$

---

[10]In the definition of $W_{\boldsymbol{A}}$, the convention is that $\lceil 0 \rceil = 1$.

**Definition B.2.** *A set $\mathcal{Q}$ of bounded measurable functions $q : [0,1] \to \mathbb{R}$ that contains $\chi$ is called a* soft affiliation model.

**Definition B.3.** *Let $D \in \mathbb{N}$. Given a soft affiliation model $\mathcal{Q}$, the subset $[\mathcal{Q}]$ of $L^2[0,1]^2 \times (L^2[0,1])^D$ of all elements of the form $(rq(x)q(y), bq(z))$, with $q \in \mathcal{Q}$, $r \in \mathbb{R}$ and $b \in \mathbb{R}^D$, is called the* soft rank-1 intersecting community graphon (ICG) model *corresponding to $\mathcal{Q}$. Given $K \in \mathbb{N}$, the subset $[\mathcal{Q}]_K$ of $L^2[0,1]^2 \times (L^2[0,1])^D$ of all linear combinations of $K$ elements of $[\mathcal{Q}]$ is called the* soft rank-$K$ ICG model *corresponding to $\mathcal{Q}$. Namely, $(C,p) \in [\mathcal{Q}]_K$ if and only if it has the form*

$$C(x,y) = \sum_{k=1}^{K} r_k q_k(x) q_k(y) \quad \text{and} \quad p(z) = \sum_{k=1}^{K} b_k q_k(z)$$

*for $(q_k)_{k=1}^{K} \in \mathcal{Q}^K$ called the* community affiliation functions, *$(r_k)_{k=1}^{K} \in \mathbb{R}^K$ called the* community affiliation magnitudes, *and $(b_k)_{k=1}^{K} \in \mathbb{R}^{K \times D}$ called the* community features.

### B.4 A constructive graphon-signal weak regularity lemma

The following theorem is the semi-constructive version of the weak regularity lemma for intersecting communities of graphons. Recall that $\alpha$ and $\beta$ denote the weights of the graphon-signal cut norm (14).

**Theorem B.1.** *Let $D \in \mathbb{N}$. Let $(W,s)$ be a $D$-channel graphon-signal, $K \in \mathbb{N}$, $\delta > 0$, and let $\mathcal{Q}$ be a soft indicators model. Let $m$ be sampled uniformly from $[K]$, and let $R \geq 1$ such that $K/R \in \mathbb{N}$. Consider the graphon-signal Frobenius norm with weights $\sqrt{\alpha \|W\|_F^2 + \beta \sum_{j=1}^{D} \|s_j\|_F^2}$. Then, in probability $1 - \frac{1}{R}$ (with respect to the choice of $m$), for every $(C^*, p^*) \in [\mathcal{Q}]_m$,*

$$\text{if} \quad \|(W,s) - (C^*, p^*)\|_F \leq (1 + \delta) \min_{(C,p) \in [\mathcal{Q}]_n} \|(W,s) - (C,p)\|_F$$

$$\text{then} \quad \|(W,s) - (C^*, p^*)\|_\square \leq$$

$$\left( \frac{3}{2} \sqrt{\alpha^2 \|W\|_F^2 + \alpha\beta \|s\|_F} + \sqrt{\alpha\beta \|W\|_F^2 + \beta^2 \|s\|_F} \right) \sqrt{\frac{R}{K}} + \delta.$$

### B.5 Proof of the soft constructive weak regularity lemma

In this subsection we prove the constructive weak graphon-signal regularity lemma.

#### B.5.1 Constructive regularity lemma in Hilbert spaces

The classical proof of the weak regularity lemma for graphons is given in [38]. The lemma is a corollary of a regularity lemma in Hilbert spaces. Our goal is to extend this lemma to be constructive. For comparison, let us first write the classical result, from [38, Lemma 4], even though we do not use this result directly.

**Lemma B.2** ([38]). *Let $\mathcal{K}_1, \mathcal{K}_2, \ldots$ be arbitrary nonempty subsets (not necessarily subspaces) of a real Hilbert space $\mathcal{H}$. Then, for every $\epsilon > 0$ and $g \in \mathcal{H}$ there is $m \leq \lceil 1/\epsilon^2 \rceil$ and $(f_i \in \mathcal{K}_i)_{i=1}^{m}$ and $(\gamma_i \in \mathbb{R})_{i=1}^{m}$, such that for every $w \in \mathcal{K}_{m+1}$*

$$\left| \left\langle w, g - \left( \sum_{i=1}^{m} \gamma_i f_i \right) \right\rangle \right| \leq \epsilon \, \|w\| \, \|g\|. \tag{16}$$

Next, we show how to modify the result to be written explicitly with an approximate minimizer of a Hilbert space norm minimization problem. The proof follows the step of [38] with some modifications required for explicitly constructing an approximate optimizer.

**Lemma B.3.** *Let $\{\mathcal{K}_j\}_{j \in \mathbb{N}}$ be a sequence of nonemply subsets of a real Hilbert space $\mathcal{H}$ and let $\delta \geq 0$. Let $K > 0$, let $R \geq 1$ such that $K/R \in \mathbb{N}$, and let $g \in \mathcal{H}$. Let $m$ be randomly uniformly sampled from $[K]$. Then, in probability $1 - \frac{1}{R}$ (with respect to the choice of $m$), any vector of the form*

$$g^* = \sum_{j=1}^{m} \gamma_j f_j \quad \text{such that} \quad \boldsymbol{\gamma} = (\gamma_j)_{j=1}^{m} \in \mathbb{R}^m \quad \text{and} \quad \mathbf{f} = (f_j)_{j=1}^{m} \in \mathcal{K}_1 \times \ldots \times \mathcal{K}_m$$

*that gives a close-to-best Hilbert space approximation of g in the sense that*

$$\|g - g^*\| \le (1 + \delta) \inf_{\boldsymbol{\gamma}, \mathbf{f}} \|g - \sum_{i=1}^{m} \gamma_i f_i\|, \tag{17}$$

*where the infimum is over $\boldsymbol{\gamma} \in \mathbb{R}^m$ and $\mathbf{f} \in \mathcal{K}_1 \times \ldots \times \mathcal{K}_m$, also satisfies*

$$\forall w \in \mathcal{K}_{m+1}, \quad |\langle w, g - g^* \rangle| \le \|w\| \, \|g\| \sqrt{\frac{R}{K} + \delta}.$$

*Proof.* Let $K > 0$. Let $R \ge 1$ such that $K/R \in \mathbb{N}$. For every $k$, let

$$\eta_k = (1 + \delta) \inf_{\boldsymbol{\gamma}, \mathbf{f}} \|g - \sum_{i=1}^{k} \gamma_i f_i\|^2$$

where the infimum is over $\boldsymbol{\gamma} = \{\gamma_1, \ldots, \gamma_k\} \in \mathbb{R}^k$ and $\mathbf{f} = \{f_1, \ldots, f_k\} \in \mathcal{K}_1 \times \ldots \times \mathcal{K}_k$. Note that $\|g\|^2 \ge \frac{\eta_1}{1+\delta} \ge \frac{\eta_2}{1+\delta} \ge \ldots \ge 0$. Therefore, there is a subset of at least $(1 - \frac{1}{R})K + 1$ indices $m$ in $[K]$ such that $\eta_m \le \eta_{m+1} + \frac{R(1+\delta)}{K} \|g\|^2$. Otherwise, there are $\frac{K}{R}$ indices $m$ in $[K]$ such that $\eta_{m+1} < \eta_m - \frac{R(1+\delta)}{K} \|g\|^2$, which means that

$$\eta_K < \eta_1 - \frac{K}{R} \frac{R(1+\delta)}{K} \|g\|^2 \le (1 + \delta) \|g\|^2 - (1 + \delta) \|g\|^2 = 0,$$

which is a contradiction to the fact that $\eta_K \ge 0$.

Hence, there is a set $\mathcal{M} \subseteq [K]$ of $(1 - \frac{1}{R})K$ indices such that for every $m \in \mathcal{M}$, every

$$g^* = \sum_{j=1}^{m} \gamma_j f_j \tag{18}$$

that satisfies

$$\|g - g^*\|^2 \le \eta_m$$

also satisfies

$$\|g - g^*\|^2 \le \eta_{m+1} + \frac{(1+\delta)R}{K} \|g\|^2. \tag{19}$$

Let $w \in \mathcal{K}_{m+1}$. By the definition of $\eta_{m+1}$, we have for every $t \in \mathbb{R}$,

$$\|g - (g^* + tw)\|^2 \ge \frac{\eta_{m+1}}{1 + \delta} \ge \frac{\|g - g^*\|^2}{1 + \delta} - \frac{R}{K} \|g\|^2.$$

This can be written as

$$\forall t \in \mathbb{R}, \quad \|w\|^2 t^2 - 2 \langle w, g - g^* \rangle t + \frac{R}{K} \|g\|^2 + (1 - \frac{1}{1+\delta}) \|g - g^*\|^2 \ge 0. \tag{20}$$

The discriminant of this quadratic polynomial is

$$4 \langle w, g - g^* \rangle^2 - 4 \|w\|^2 \left( \frac{R}{K} \|g\|^2 + (1 - \frac{1}{1+\delta}) \|g - g^*\|^2 \right)$$

and it must be non-positive to satisfy the inequality (20), namely

$$4 \langle w, g - g^* \rangle^2 \le 4 \|w\|^2 \left( \frac{R}{K} \|g\|^2 + (1 - \frac{1}{1+\delta}) \|g - g^*\|^2 \right)$$

$$\le 4 \|w\|^2 \left( \frac{R}{K} \|g\|^2 + (1 - \frac{1}{1+\delta}) \eta_m \right)$$

$$\le 4 \|w\|^2 \left( \frac{R}{K} \|g\|^2 + (1 - \frac{1}{1+\delta})(1 + \delta) \|g\|^2 \right)$$

which proves

$$\langle w, g - g^* \rangle \le \|w\| \, \|g\| \sqrt{\frac{R}{K} + \delta}.$$

This is also true for $-w$, which concludes the proof. $\qquad \square$

### B.5.2 Proof of the Soft Constructive Weak Regularity Lemma for Graphon-Signals

For two functions $f, g : [0, 1] \to \mathbb{R}$, we denote by $f \otimes g : [0, 1]^2 \to \mathbb{R}$ the function

$$f \otimes g(x, y) = f(x)g(y).$$

We recall here Theorem B.1 for the convenience of the reader.

**Theorem B.1.** *Let $(W, s)$ be a graphon-signal, $K \in \mathbb{N}$, $\delta > 0$, and let $\mathcal{Q}$ be a soft indicators model. Let $m$ be sampled uniformly from $[K]$, and let $R \geq 1$ such that $K/R \in \mathbb{N}$. Consider the graphon-signal Frobenius norm with weights $\sqrt{\alpha \|W\|_F^2 + \beta \sum_{j=1}^D \|s_j\|_F^2}$. Then, in probability $1 - \frac{1}{R}$ (with respect to the choice of $m$), for every $(C^*, p^*) \in [\mathcal{Q}]_m$,*

*if* $\quad \|(W, s) - (C^*, p^*)\|_F \leq (1 + \delta) \min_{(C,p) \in [\mathcal{Q}]_n} \|(W, s) - (C, p)\|_F$

*then* $\quad \|(W, s) - (C^*, p^*)\|_\square \leq$

$$\left( \frac{3}{2} \sqrt{\alpha^2 \|W\|_F^2 + \alpha\beta \|s\|_F} + \sqrt{\alpha\beta \|W\|_F^2 + \beta^2 \|s\|_F} \right) \sqrt{\frac{R}{K} + \delta}.$$

The proof follows the techniques of a part of the proof of the weak graphon regularity lemma from [38], while tracking the approximate minimizer in our formulation of Lemma B.3. This requires a probabilistic setting, and extending to soft indicators models. We note that the weak regularity lemma in [38] is formulated for non-intersecting blocks, but the intersecting community version is an intermediate step in its proof.

*Proof.* Let us use Lemma B.3, with $\mathcal{H} = L^2[0, 1]^2 \times (L^2[0, 1])^D$ with the weighted inner product

$$\langle (W, s), (W', s') \rangle = \alpha \iint_{[0,1]^2} W(x, y)W'(x, y) dx dy + \beta \sum_{j=1}^D \int_{[0,1]} s_j(x) s'_j(x) dx,$$

and corresponding norm $\sqrt{\alpha \|W\|_F^2 + \beta \sum_{j=1}^D \|s_j\|_F^2}$ and corresponding weighted inner product, and $\mathcal{K}_j = [\mathcal{Q}]$. Note that the Hilbert space norm is the Frobenius norm in this case. In the setting of the lemma, we take $g = (W, s)$, and $g^* \in [\mathcal{Q}]_m$. By the lemma, in the event of probability $1 - 1/R$, any approximate Frobenius minimizer $(C^*, p^*)$, namely, that satisfies $\|(W, s) - (C^*, p^*)\|_F \leq (1 + \delta) \min_{(C,p) \in [\mathcal{Q}]_m} \|(W, s) - (C, p)\|_F$, also satisfies

$$\langle (T, y), (W, s) - (C^*, p^*) \rangle \leq \|(T, y)\|_F \|(W, s)\|_F \sqrt{\frac{R}{K} + \delta}$$

for every $(T, y) \in [\mathcal{Q}]$. Hence, for every choice of measurable subsets $\mathcal{S}, \mathcal{T} \subset [0, 1]$, we have

$$\left| \int_{\mathcal{S}} \int_{\mathcal{T}} (W(x, y) - C^*(x, y)) dx dy \right|$$

$$= \frac{1}{2} \left| \int_{\mathcal{S}} \int_{\mathcal{T}} (W(x, y) - C^*(x, y)) dx dy + \int_{\mathcal{T}} \int_{\mathcal{S}} (W(x, y) - C^*(x, y)) dx dy \right|$$

$$= \frac{1}{2} \left| \int_{\mathcal{S} \cup \mathcal{T}} \int_{\mathcal{S} \cup \mathcal{T}} (W(x, y) - C^*(x, y)) dx dy - \int_{\mathcal{S}} \int_{\mathcal{S}} (W(x, y) - C^*(x, y)) dx dy \right.$$

$$\left. - \int_{\mathcal{T}} \int_{\mathcal{T}} (W(x, y) - C^*(x, y)) dx dy \right|$$

$$\leq \left| \frac{1}{2\alpha} \langle (\mathbb{1}_{\mathcal{S} \cup \mathcal{T}} \otimes \mathbb{1}_{\mathcal{S} \cup \mathcal{T}}, 0), (W, s) - (C^*, p^*) \rangle \right| + \left| \frac{1}{2\alpha} \langle (\mathbb{1}_{\mathcal{S}} \otimes \mathbb{1}_{\mathcal{S}}, 0), (W, s) - (C^*, p^*) \rangle \right|$$

$$+ \left| \frac{1}{2\alpha} \langle (\mathbb{1}_{\mathcal{T}} \otimes \mathbb{1}_{\mathcal{T}}, 0), (W, s) - (C^*, p^*) \rangle \right|$$

$$\leq \frac{1}{2\alpha} \|(W, s)\|_F \left( \|(\mathbb{1}_{\mathcal{S} \cup \mathcal{T}} \otimes \mathbb{1}_{\mathcal{S} \cup \mathcal{T}}, 0)\|_F + \|(\mathbb{1}_{\mathcal{S}} \otimes \mathbb{1}_{\mathcal{S}}, 0)\|_F + \|(\mathbb{1}_{\mathcal{T}} \otimes \mathbb{1}_{\mathcal{T}}, 0)\|_F \right) \sqrt{\frac{R}{K} + \delta}$$

$$\leq \frac{3}{2\alpha} \sqrt{\alpha \|W\|_F^2 + \beta \|s\|_F} \sqrt{\alpha} \sqrt{\frac{R}{K} + \delta}$$

Hence, taking the supremum over $\mathcal{S}, \mathcal{T} \subset [0,1]$, we also have

$$\alpha \|W - C^*\|_\square \leq \frac{3}{2} \sqrt{\alpha^2 \|W\|_{\mathrm{F}}^2 + \alpha\beta \|s\|_{\mathrm{F}}} \sqrt{\frac{R}{K} + \delta}.$$

Similarly, for every measurable $\mathcal{T} \subset [0,1]$ and every standard basis element $\boldsymbol{b} = (\delta_{j,i})_{i=1}^D$ for any $j \in [D]$,

$$
\begin{aligned}
&\left| \int_{\mathcal{T}} (s_j(x) - p_j^*(x)) dx \right| \\
&= \left| \frac{1}{\beta} \langle (0, \boldsymbol{b} \mathbb{1}_{\mathcal{T}}), (W, s) - (C^*, p^*) \rangle \right| \\
&\leq \frac{1}{\beta} \|(W, s)\|_{\mathrm{F}} \|(0, \boldsymbol{b} \mathbb{1}_{\mathcal{T}})\|_{\mathrm{F}} \sqrt{\frac{R}{K} + \delta} \\
&\leq \frac{1}{\beta} \sqrt{\alpha \|W\|_{\mathrm{F}}^2 + \beta \|s\|_{\mathrm{F}}} \sqrt{\beta} \sqrt{\frac{R}{K} + \delta},
\end{aligned}
$$

so, taking the supremum over $\mathcal{T} \subset [0,1]$ independently for every $j \in [D]$, and averaging over $j \in [D]$, we get

$$\beta \|s - p^*\|_\square \leq \sqrt{\alpha\beta \|W\|_{\mathrm{F}}^2 + \beta^2 \|s\|_{\mathrm{F}}} \sqrt{\frac{R}{K} + \delta}.$$

Overall, we get

$$\|(W, s) - (C^*, p^*)\|_\square \leq \left( \frac{3}{2} \sqrt{\alpha^2 \|W\|_{\mathrm{F}}^2 + \alpha\beta \|s\|_{\mathrm{F}}} + \sqrt{\alpha\beta \|W\|_{\mathrm{F}}^2 + \beta^2 \|s\|_{\mathrm{F}}} \right) \sqrt{\frac{R}{K} + \delta}.$$

$\square$

### B.6 Proof of the constructive weak regularity lemma for sparse graph-signals

We now show that Theorem 3.1 is a special case of Theorem B.1, restricted to graphon-signals induced by graph-signals, up to choice of the graphon-signal weights $\alpha$ and $\beta$. Namely, choose $\alpha = \alpha' N^2 / E$ and $\beta = \beta'$ with $\alpha' + \beta' = 1$ as the weights of the graphon-signal cut norm of Theorem B.1, with arbitrary $\alpha', \beta' \geq 0$ satisfying $\alpha' + \beta' = 1$. It is easy to see the following relation between the graphon-signal and graph-signal cut norms

$$\|(W_{\boldsymbol{A}}, s_{\boldsymbol{S}}) - (W_{\boldsymbol{C}^*}, s_{\boldsymbol{P}^*})\|_\square = \|(\boldsymbol{A}, \boldsymbol{S}) - (\boldsymbol{C}^*, \boldsymbol{P}^*)\|_{\square, N, E},$$

where $\|(\boldsymbol{A}, \boldsymbol{S}) - (\boldsymbol{C}^*, \boldsymbol{P}^*)\|_{\square, N, E}$ is based on the weights $\alpha'$ and $\beta'$. Now, since $\deg(\boldsymbol{A})/N^2 = \|\boldsymbol{A}\|_{\mathrm{F}}^2 = \frac{E}{N^2}$, and by $\|\boldsymbol{S}\|_{\mathrm{F}} \leq 1$, the bound in Theorem B.1 becomes

$$\|(\boldsymbol{A}, \boldsymbol{S}) - (\boldsymbol{C}^*, \boldsymbol{P}^*)\|_{\square, N, E} \leq \left( \frac{3}{2} \sqrt{\alpha'^2 \frac{N^4}{E^2} \frac{E}{N^2} + \alpha'\beta' \frac{N^2}{E}} + \sqrt{\alpha'\beta' \frac{N^2}{E} \frac{E}{N^2} + \beta'^2} \right) \sqrt{\frac{R}{K} + \delta}$$

$$\leq C \frac{N}{\sqrt{E}} \sqrt{\frac{R}{K} + \delta}$$

where, since $\alpha' + \beta' = 1$, and by convexity of the square root,

$$C = \frac{3}{2}\sqrt{\alpha'} + \sqrt{\beta'} \leq \frac{3}{2}\sqrt{\alpha'} + \frac{3}{2}\sqrt{\beta'} \leq \frac{3}{2}\sqrt{\alpha' + \beta'} = \frac{3}{2}.$$

The only thing left to do is to change the notations $\alpha' \mapsto \alpha$ and $\beta' \mapsto \beta$.

## C   Fitting ICGs to graphs efficiently

Here, we prove Proposition 4.1. For convenience, we recall the proposition.

**Proposition 4.1.** *Let $\boldsymbol{A} = (a_{i,j})_{i,j=1}^N$ be an adjacency matrix of a weighted graph with $E$ edges. The graph part of the Frobenius loss can be written as*

$$\left\| \boldsymbol{A} - \boldsymbol{Q}\operatorname{diag}(\boldsymbol{r})\boldsymbol{Q}^\top \right\|_{\mathrm{F}}^2 = \frac{1}{N^2} \operatorname{Tr}\left( (\boldsymbol{Q}^\top\boldsymbol{Q})\operatorname{diag}(\boldsymbol{r})(\boldsymbol{Q}^\top\boldsymbol{Q})\operatorname{diag}(\boldsymbol{r}) \right) + \|\boldsymbol{A}\|_{\mathrm{F}}^2$$

$$- \frac{2}{N^2} \sum_{i=1}^{N} \sum_{j \in \mathcal{N}(i)} \boldsymbol{Q}_{i,:}\operatorname{diag}(\boldsymbol{r}) \left(\boldsymbol{Q}^\top\right)_{:,j} a_{i,j}.$$

*Computing the right-hand-side and its gradients with respect to $\boldsymbol{Q}$ and $\boldsymbol{r}$ has a time complexity of $\mathcal{O}(K^2 N + KE)$, and a space complexity of $\mathcal{O}(KN + E)$.*

*Proof.* The loss can be expressed as

$$\left\| \boldsymbol{A} - \boldsymbol{Q}\operatorname{diag}(\boldsymbol{r})\boldsymbol{Q}^\top \right\|_{\mathrm{F}}^2 = \frac{1}{N^2} \sum_{i,j=1}^{N} \left( a_{i,j} - \boldsymbol{Q}_{i,:}\operatorname{diag}(\boldsymbol{r}) \left(\boldsymbol{Q}^\top\right)_{:,j} \right)^2$$

$$= \frac{1}{N^2} \sum_{i=1}^{N} \left( \sum_{j \in \mathcal{N}(i)} \left( a_{i,j} - \boldsymbol{Q}_{i,:}\operatorname{diag}(\boldsymbol{r}) \left(\boldsymbol{Q}^\top\right)_{:,j} \right)^2 + \sum_{j \notin \mathcal{N}(i)} \left( -\boldsymbol{Q}_{i,:}\operatorname{diag}(\boldsymbol{r}) \left(\boldsymbol{Q}^\top\right)_{:,j} \right)^2 \right)$$

$$= \frac{1}{N^2} \sum_{i=1}^{N} \left( \sum_{j \in \mathcal{N}(i)} \left( a_{i,j} - \boldsymbol{Q}_{i,:}\operatorname{diag}(\boldsymbol{r}) \left(\boldsymbol{Q}^\top\right)_{:,j} \right)^2 + \right.$$

$$+ \sum_{j=1}^{N} \left( \boldsymbol{Q}_{i,:}\operatorname{diag}(\boldsymbol{r}) \left(\boldsymbol{Q}^\top\right)_{:,j} \right)^2 - \sum_{j \in \mathcal{N}(i)} \left( \boldsymbol{Q}_{i,:}\operatorname{diag}(\boldsymbol{r}) \left(\boldsymbol{Q}^\top\right)_{:,j} \right)^2 \left. \right)$$

We expand the quadratic term $\left( a_{i,j} - \boldsymbol{Q}_{i,:}\operatorname{diag}(\boldsymbol{r}) \left(\boldsymbol{Q}^\top\right)_{:,j} \right)^2$, and get

$$\left\| \boldsymbol{A} - \boldsymbol{Q}\operatorname{diag}(\boldsymbol{r})\boldsymbol{Q}^\top \right\|_{\mathrm{F}}^2$$

$$= \frac{1}{N^2} \sum_{i,j=1}^{N} \left( \boldsymbol{Q}_{i,:}\operatorname{diag}(\boldsymbol{r}) \left(\boldsymbol{Q}^\top\right)_{:,j} \right)^2 + \frac{1}{N^2} \sum_{i=1}^{N} \sum_{j \in \mathcal{N}(i)} \left( a_{i,j}^2 - 2\boldsymbol{Q}_{i,:}\operatorname{diag}(\boldsymbol{r}) \left(\boldsymbol{Q}^\top\right)_{:,j} a_{i,j} \right)$$

$$= \left\| \boldsymbol{Q}\operatorname{diag}(\boldsymbol{r})\boldsymbol{Q}^\top \right\|_{\mathrm{F}}^2 + \frac{1}{N^2} \sum_{i,j=1}^{N} a_{i,j}^2 - \frac{2}{N^2} \sum_{i=1}^{N} \sum_{j \in \mathcal{N}(i)} \boldsymbol{Q}_{i,:}\operatorname{diag}(\boldsymbol{r}) \left(\boldsymbol{Q}^\top\right)_{:,j} a_{i,j}$$

$$= \frac{1}{N^2} \operatorname{Tr}\left( \boldsymbol{Q}\operatorname{diag}(\boldsymbol{r})\boldsymbol{Q}^\top\boldsymbol{Q}\operatorname{diag}(\boldsymbol{r})\boldsymbol{Q}^\top \right) + \frac{1}{N^2} \sum_{i,j=1}^{N} a_{i,j}^2$$

$$- \frac{2}{N^2} \sum_{i=1}^{N} \sum_{j \in \mathcal{N}(i)} \boldsymbol{Q}_{i,:}\operatorname{diag}(\boldsymbol{r}) \left(\boldsymbol{Q}^\top\right)_{:,j} a_{i,j}$$

$$= \frac{1}{N^2} \operatorname{Tr}\left( \boldsymbol{Q}^\top\boldsymbol{Q}\operatorname{diag}(\boldsymbol{r})\boldsymbol{Q}^\top\boldsymbol{Q}\operatorname{diag}(\boldsymbol{r}) \right) + \frac{1}{N^2} \sum_{i,j=1}^{N} a_{i,j}^2$$

$$- \frac{2}{N^2} \sum_{i=1}^{N} \sum_{j \in \mathcal{N}(i)} \boldsymbol{Q}_{i,:}\operatorname{diag}(\boldsymbol{r}) \left(\boldsymbol{Q}^\top\right)_{:,j} a_{i,j}$$

The last equality follows the identity $\forall \boldsymbol{B}, \boldsymbol{D} \in \mathbb{R}^{N \times K} : \operatorname{Tr}(\boldsymbol{B}\boldsymbol{D}^\top) = \operatorname{Tr}(\boldsymbol{D}^\top\boldsymbol{B})$, with $\boldsymbol{B} = \boldsymbol{Q}\operatorname{diag}(\boldsymbol{r})\boldsymbol{Q}^\top\boldsymbol{Q}\operatorname{diag}(\boldsymbol{r})$ and $\boldsymbol{D} = \boldsymbol{Q}^\top$.

The middle term in the last line is constant and can thus be omitted in the optimization process. The leftmost term in the last line is calculated by performing matrix multiplication from right to left, or by first computing $\boldsymbol{Q}^\top\boldsymbol{Q}$ and then the rest of the product. Thus, the time complexity is

$\mathcal{O}(K^2N)$ and the largest matrix held in memory is of size $\mathcal{O}(KN)$. The rightmost term is calculated by message-passing, which has time complexity of $\mathcal{O}(KE)$. Thus, Computing the right-hand-side and its gradients with respect to $\boldsymbol{Q}$ and $\boldsymbol{r}$ has a time complexity of $\mathcal{O}(K^2N + KE)$ and a space complexity of $\mathcal{O}(KN + E)$. $\qquad\square$

## D   Initializing the optimization with eigenvectors

Next, we propose a good initialization for the GD minimization of (7). For that, we consider a corollary of the constructive soft weak regularity for grapohns, with the signal weight in the cut norm set to $\beta = 0$, using all $L^2[0,1]$ functions as the soft affiliation model, and taking relative Frobenius error with $\delta = 0$. In this case, the best rank-$K$ approximation theorem (Eckart–Young–Mirsky Theorem [51, Theorem 5.9]), the minimizer of the Frobenius error is the projection of $W$ upon its leading eigenvectors. This leads to the following corollary.

**Corollary D.1.** *Let $\boldsymbol{A} \in [0,1]^{N \times N}$ be a matrix with $\deg(\boldsymbol{A}) = N^2 \|\boldsymbol{A}\|_{\mathrm{F}}^2 = E'$. Let $K \in \mathbb{N}$, let $m$ be sampled uniformly from $[K]$, and let $R \geq 1$ such that $K/R \in \mathbb{N}$. Let $\phi_1, \ldots, \phi_m$ be the leading eigenvectors of $\boldsymbol{A}$, with eigenvalues $\lambda_1, \ldots, \lambda_m$ of highest magnitudes $|\lambda_1| \geq |\lambda_2| \geq \ldots \geq |\lambda_m|$, and let $\boldsymbol{C}^* = \sum_{k=1}^{m} \lambda_k \phi_k \phi_k^\top$. Then, in probability $1 - \frac{1}{R}$ (with respect to the choice of $m$),*

$$\|\boldsymbol{A} - \boldsymbol{C}^*\|_{\square;N,E'} \leq \frac{3N}{2\sqrt{E'}} \sqrt{\frac{R}{K}}.$$

The initialization is based on Corollary D.1 as follows. We consider the leading $K/3$ eigenvectors, assuming $K$ is divisible by 3, for reasons that will become clear shortly. Hence, for $\boldsymbol{C} = \boldsymbol{\Phi}_{K/3} \boldsymbol{\Lambda}_{K/3} \boldsymbol{\Phi}_{K/3}^\mathrm{T}$, where $\boldsymbol{\Phi}_{K/3} \in \mathbb{R}^{N \times K/3}$ is the matrix consisting of the leading eigenvectors of $\boldsymbol{A}$ as columns, and $\boldsymbol{\Lambda}_{K/3} \in \mathbb{R}^{K/3 \times K/3}$ the diagonal matrix of the leading eigenvalues, we have

$$\|\boldsymbol{A} - \boldsymbol{C}\|_{\square;N,E} < \frac{3N}{2\sqrt{E'}} \sqrt{\frac{3R}{K}}. \tag{21}$$

To obtain soft affiliations with values in $[0, 1]$, we now initialize $\boldsymbol{Q}$ as described in (8).

## E   Learning ICG with subgraph SGD

In this section we prove that the gradients of the subgraph SGD approximate the gradients of the full GD method.

**Proposition E.1.** *Let $0 < p < 1$. Under the above setting, if we restrict all entries of $\boldsymbol{C}$, $\boldsymbol{P}$, $\boldsymbol{Q}$, $\boldsymbol{r}$ and $\boldsymbol{F}$ to be in $[0, 1]$, then in probability at least $1 - p$, for every $k \in [K]$, $d \in [D]$ and $m \in [M]$*

$$\left| \nabla_{q_{n_m,k}} L - \frac{M}{N} \nabla_{q_{n_m,k}} L^{(\boldsymbol{n})} \right| \leq \frac{4}{N} \sqrt{\frac{2\log(1/p_3) + 2\log(N) + 2\log(K) + 2\log(2)}{M}},$$

$$\left| \nabla_{r_k} L - \nabla_{r_k} L^{(\boldsymbol{n})} \right| \leq 4 \sqrt{\frac{2\log(1/p_1) + 2\log(N) + 2\log(K) + 2\log(2)}{M}},$$

$$\left| \nabla_{f_{k,d}} L - \nabla_{f_{k,d}} L^{(\boldsymbol{n})} \right| \leq \frac{4\lambda}{D} \sqrt{\frac{2\log(1/p_2) + 2\log(K) + 2\log(D) + 2\log(2)}{M}}.$$

Proposition E.1 means that the gradient with respect to $\boldsymbol{r}$ and $\boldsymbol{F}$ computed at each SGD step approximate the full GD gradients. The gradients with respect to $\boldsymbol{Q}$ approximate a scaled version of the full gradients, and only on the sampled nodes, where the unsampled nodes are not updated in the SGD step. This means that when optimizing the ICG with SGD, we need to scale the gradients with resect to $\boldsymbol{Q}$ of the loss by $\frac{M}{N}$. To use the same entry-wise learning rate in SGD as in GD one must multiply the loss (9) by $M/N$. Hence, SGD needs in expectation $N/M$ times the number of steps that GD requires. This means that in SGD we trade the memory complexity (reducing it by a factor $M/N$) by time complexity (increasing it by a factor $N/M$). Note that slower run-time, while not desirable, can be tolerated, while higher memory requirement is often prohibitive due to hardware limitations.

From here until the end of this section we prove Proposition E.1. For that, we compute the gradients of the full loss $L$ and the sampled loss $L^{(n)}$.

To avoid tensor notations, we treat by abuse of notation the gradient of a function as a linear operator. Recall that the differential $\mathcal{D}_r \boldsymbol{T} = \mathcal{D}_r \boldsymbol{T}(\boldsymbol{r}')$ of the function $\boldsymbol{T} : \mathbb{R}^R \to \mathbb{R}^U$ at the point $\boldsymbol{r}' \in \mathbb{R}^R$ is the unique linear operator $\mathbb{R}^R \to \mathbb{R}^U$ that describes the linearization of $\boldsymbol{T}$ about $\boldsymbol{r}'$ via

$$\boldsymbol{T}(\boldsymbol{r}' + \epsilon \boldsymbol{r}'') = \boldsymbol{T}(\boldsymbol{r}') + \epsilon \mathcal{D}_r \boldsymbol{T} \boldsymbol{r}'' + o(\epsilon)$$

where $\epsilon \in \mathbb{R}$. Given an inner product in $\mathbb{R}^R$, the gradient $\nabla_r \boldsymbol{T} = \nabla_r \boldsymbol{T}(\boldsymbol{r}') \in \mathbb{R}^{U \times R}$ of the function $\boldsymbol{T} : \mathbb{R}^R \to \mathbb{R}^U$ at the point $\boldsymbol{r}' \in \mathbb{R}^R$ is defined to be the vector (guaranteed to uniquely exist by the Riesz representation theorem) that satisfies

$$\mathcal{D}_r \boldsymbol{T} \boldsymbol{r}'' = \langle \boldsymbol{r}'', \nabla_r \boldsymbol{T} \rangle$$

for every $\boldsymbol{r}'' \in \mathbb{R}^R$. Here, the inner product if defined row-wise by

$$\langle \boldsymbol{r}'', \nabla_r \boldsymbol{T} \rangle := \left( \langle \boldsymbol{r}'', \nabla_r \boldsymbol{T}[u, :] \rangle \right)_{u=1}^U,$$

where $\boldsymbol{T}[u, :]$ is the $u$-th row of $\nabla_r \boldsymbol{T}$. In our analysis, by abuse of notation, we identify $\nabla_r \boldsymbol{T}$ with $\mathcal{D}_r \boldsymbol{T}$ for a fixed given inner product.

Define the inner product between matrices

$$\langle \boldsymbol{B}, \boldsymbol{D} \rangle_2 = \sum_{i,j} b_{i,j} d_{i,j}.$$

Denote

$$L(\boldsymbol{C}, \boldsymbol{P}) = \|\boldsymbol{A} - \boldsymbol{C}\|_{\mathrm{F}}^2 + \lambda \|\boldsymbol{S} - \boldsymbol{P}\|_{\mathrm{F}}^2.$$

First, the gradient of $\|\boldsymbol{A} - \boldsymbol{C}\|_{\mathrm{F}}^2$ with respect to $\boldsymbol{C}$ is

$$\nabla_C \|\boldsymbol{A} - \boldsymbol{C}\|_{\mathrm{F}}^2 = -\frac{2}{N^2}(\boldsymbol{A} - \boldsymbol{C}),$$

which is identified by abuse of notation by the linear functional $\mathbb{R}^{N^2} \to \mathbb{R}$

$$\boldsymbol{D} \mapsto \nabla_C \|\boldsymbol{A} - \boldsymbol{C}\|_{\mathrm{F}}^2 (\boldsymbol{D}) = \left\langle \boldsymbol{D}, -\frac{2}{N^2}(\boldsymbol{A} - \boldsymbol{C}) \right\rangle_2.$$

Similarly,

$$\nabla_P \|\boldsymbol{S} - \boldsymbol{P}\|_{\mathrm{F}}^2 = -\frac{2}{ND}(\boldsymbol{S} - \boldsymbol{P}) : \mathbb{R}^{ND} \to \mathbb{R}.$$

Given any parameter $\boldsymbol{z} \in \mathbb{R}^R$ on which the graph $\boldsymbol{C} = \boldsymbol{C}(\boldsymbol{z})$ and the signal $\boldsymbol{P} = \boldsymbol{P}(\boldsymbol{z})$ depend, we have

$$\nabla_z L(\boldsymbol{C}, \boldsymbol{P}) = \frac{2}{N^2}(\boldsymbol{C} - \boldsymbol{A})\nabla_z \boldsymbol{C} + \lambda \frac{2}{ND}(\boldsymbol{P} - \boldsymbol{S})\nabla_z \boldsymbol{P}, \tag{22}$$

where $\nabla_z \boldsymbol{C} : \mathbb{R}^R \to \mathbb{R}^{N^2}$ and $\nabla_z \boldsymbol{C} \in \mathbb{R}^R \to \mathbb{R}^{ND}$ are linear operators, and the products in (22) are operator composition, namely, in coordinates it is elementwise multiplication (not matrix multiplication).

Let us now compute the gradients of $\boldsymbol{C} = \boldsymbol{Q} \operatorname{diag}(\boldsymbol{r}) \boldsymbol{Q}^\top$ and $\boldsymbol{P} = \boldsymbol{Q} \boldsymbol{F}$ with respect to $\boldsymbol{Q}, \boldsymbol{r}$ and $\boldsymbol{F}$ in coordinates. We have

$$\nabla_{r_k} c_{i,j} = q_{i,k} q_{j,k},$$

and

$$\nabla_{q_{m,k}} c_{i,j} = r_k \delta_{i-m} q_{j,k} + r_k \delta_{j-m} q_{i,k},$$

where $\delta_l$ is 1 if $l = 0$ and zero otherwise. Moreover,

$$\nabla_{f_{k,l}} p_{i,d} = q_{i,k} \delta_{l-d},$$

and

$$\nabla_{q_{m,k}} p_{i,d} = f_{k,d} \delta_{i-m}.$$

Hence,

$$\nabla_{r_k} L = \frac{2}{N^2} \sum_{i,j=1}^{N} (c_{i,j} - a_{i,j}) q_{i,k} q_{j,k},$$

$$\nabla_{q_{m,k}} L = \frac{2}{N^2} \sum_{j=1}^{N} (c_{m,j} - a_{m,j}) r_k q_{j,k} + \frac{2}{N^2} \sum_{i=1}^{N} (c_{i,m} - a_{i,m}) r_k q_{i,k} + \lambda \frac{2}{ND} \sum_{d=1}^{D} (p_{m,d} - s_{m,d}) f_{k,d},$$

and

$$\nabla_{f_{k,l}} L = \lambda \frac{2}{ND} \sum_{i=1}^{N} (p_{i,l} - s_{i,l}) q_{i,k}.$$

Similarly,

$$\nabla_{r_k} L^{(\boldsymbol{n})} = \frac{2}{M^2} \sum_{i,j=1}^{M} (c_{n_i,n_j} - a_{n_i,n_j}) q_{n_i,k} q_{n_j,k},$$

$$\nabla_{q_{n_m,k}} L^{(\boldsymbol{n})} = \frac{2}{M^2} \sum_{j=1}^{M} (c_{n_m,n_j} - a_{n_m,n_j}) c_k q_{n_j,k}$$

$$+ \frac{2}{M^2} \sum_{i=1}^{M} (c_{n_i,n_m} - a_{n_i,n_m}) c_k q_{n_i,k} + \lambda \frac{2}{MD} \sum_{d=1}^{D} (p_{n_m,d} - s_{n_m,d}) f_{k,d},$$

and

$$\nabla_{f_{k,l}} L^{(\boldsymbol{n})} = \lambda \frac{2}{MD} \sum_{i=1}^{M} (p_{n_i,l} - s_{n_i,l}) q_{n_i,k}.$$

We next derive the convergence analysis, based on Hoeffding's Inequality and two Monte-Carlo approximation lemmas.

**Theorem E.2** (Hoeffding's Inequality). *Let $Y_1, \ldots, Y_M$ be independent random variables such that $a \leq Y_m \leq b$ almost surely. Then, for every $k > 0$,*

$$\mathbb{P}\Big( \Big| \frac{1}{M} \sum_{m=1}^{M} (Y_m - \mathbb{E}[Y_m]) \Big| \geq k \Big) \leq 2 \exp\Big( - \frac{2k^2 M}{(b-a)^2} \Big).$$

The following is a standard Monte Carlo approximation error bound based on Hoeffding's inequality.

**Lemma E.3.** *Let $\{i_m\}_{m=1}^{M}$ be uniform i.i.d in $[N]$. Let $\boldsymbol{v} \in \mathbb{R}^N$ be a vector with entries $v_n$ in the set $[-1,1]$. Then, for every $0 < p < 1$, in probability at least $1 - p$*

$$\Big| \frac{1}{M} \sum_{m=1}^{M} v_{i_m} - \frac{1}{N} \sum_{n=1}^{N} v_n \Big| \leq \sqrt{\frac{2\log(1/p) + 2\log(2)}{M}}.$$

*Proof.* This is a direct result of Hoeffding's Inequality on the i.i.d. variables $\{v_{i_m}\}_{m=1}^{M}$. $\qquad \square$

The next lemma derives a Monte Carlo approximation error bound based on Hoeffding's inequality for 2D arrays, in case one samples only 1D independent sample points, and the 2D sample points are all pairs of the 1D points (which are hence no longer independent).

**Lemma E.4.** *Let $\{i_m\}_{m=1}^{M}$ be uniform i.i.d in $[N]$. Let $\boldsymbol{A} \in \mathbb{R}^{N \times N}$ be symmetric with $a_{i,j} \in [-1,1]$. Then, for every $0 < p < 1$, in probability more than $1 - p$*

$$\Big| \frac{1}{N^2} \sum_{j=1}^{N} \sum_{n=1}^{N} a_{j,n} - \frac{1}{M^2} \sum_{m=1}^{M} \sum_{l=1}^{M} a_{i_m,i_l} \Big| \leq 2\sqrt{\frac{2\log(1/p) + 2\log(N) + 2\log(2)}{M}}.$$

*Proof.* Let $0 < p < 1$. For each fixed $n \in [N]$, consider the independent random variables $Y_m^n = a_{i_m, n}$, with

$$\mathbb{E}(Y_m^n) = \frac{1}{N} \sum_{j=1}^{N} a_{j,n},$$

and $-1 \leq Y_m \leq 1$. By Hoeffding's Inequality, for $k = \sqrt{\frac{2 \log(1/p) + 2 \log(N) + 2 \log(2)}{M}}$, we have

$$\left| \frac{1}{N} \sum_{j=1}^{N} a_{j,n} - \frac{1}{M} \sum_{m=1}^{M} a_{i_m, n} \right| \leq k$$

in an event $\mathcal{E}_n$ of probability more than $1 - p/N$. Intersecting the events $\{\mathcal{E}_n\}_{n=1}^{N}$, we get

$$\forall n \in [N] : \quad \left| \frac{1}{N} \sum_{j=1}^{N} a_{j,n} - \frac{1}{M} \sum_{m=1}^{M} a_{i_m, n} \right| \leq k$$

in the event $\mathcal{E} = \cap_n \mathcal{E}_n$ probability at least $1 - p$. Hence, by the triangle inequality, we also have in the event $\mathcal{E}$

$$\left| \frac{1}{NM} \sum_{l=1}^{M} \sum_{j=1}^{N} a_{j,i_l} - \frac{1}{M^2} \sum_{l=1}^{M} \sum_{m=1}^{M} a_{i_m, i_l} \right| \leq k,$$

and

$$\left| \frac{1}{N^2} \sum_{n=1}^{N} \sum_{j=1}^{N} a_{j,n} - \frac{1}{NM} \sum_{n=1}^{N} \sum_{m=1}^{M} a_{i_m, n} \right| \leq k.$$

Hence, by the symmetry of $\boldsymbol{A}$ and by the triangle inequality,

$$\left| \frac{1}{N^2} \sum_{n=1}^{N} \sum_{j=1}^{N} a_{j,n} - \frac{1}{M^2} \sum_{l=1}^{M} \sum_{m=1}^{M} a_{i_m, i_l} \right| \leq 2k.$$

$\square$

Now, since all entries of $\boldsymbol{A}$, $\boldsymbol{S}$, $\boldsymbol{C}$, $\boldsymbol{P}$, $\boldsymbol{Q}$, $\boldsymbol{r}$ and $\boldsymbol{F}$ are in $[0, 1]$, we may use Lemmas E.3 and E.4 to derive approximation errors for the gradients of $L$. Specifically, for any $0 < p_1 < 1$, for every $k \in [K]$ there is an event $\mathcal{A}_k$ of probability at least $1 - p_1$ such that

$$\left| \nabla_{r_k} L - \nabla_{r_k} L^{(\boldsymbol{n})} \right| \leq 4 \sqrt{\frac{2 \log(1/p_1) + 2 \log(N) + 2 \log(2)}{M}}.$$

Moreover, for every $k \in [K]$ and $j \in [D]$, and every $0 < p_2 < 1$ there is an event $\mathcal{C}_{k,j}$ of probability at least $1 - p_2$ such that

$$\left| \nabla_{f_{k,l}} L - \nabla_{f_{k,l}} L^{(\boldsymbol{n})} \right| \leq \frac{4\lambda}{D} \sqrt{\frac{2 \log(1/p_2) + 2 \log(2)}{M}}.$$

For the approximation analysis of $\nabla_{q_{n_m, l}} L$, note that the index $n_m$ is random, so we derive a uniform convergence analysis for all possible values of $n_m$. For that, for every $n \in [N]$ and $k \in [K]$, define the vector

$$\widetilde{\nabla_{q_{n,k}} L^{(\boldsymbol{n})}} = \frac{2}{M^2} \sum_{j=1}^{M} (c_{n,n_j} - a_{n,n_j}) c_k q_{n_j, k}$$

$$+ \frac{2}{M^2} \sum_{i=1}^{M} (c_{n_i, n} - a_{n_i, n}) c_k q_{n_i, k} + \lambda \frac{2}{MD} \sum_{d=1}^{D} (p_{n,d} - s_{n,d}) f_{k,d},$$

Note that $\widetilde{\nabla_{q_{n,k}} L^{(\boldsymbol{n})}}$ is not a gradient of $L^{(\boldsymbol{n})}$ (since if $n$ is not a sample from $\{n_m\}$ the gradient must be zero), but is denoted with $\nabla$ for its structural similarity to $\nabla_{q_{n_m, k}} L^{(\boldsymbol{n})}$. Let $0 < p_3 < 1$. By

Lemma E.3, for every $k \in [K]$ there is an event $\mathcal{B}_k$ of probability at least $1 - p_3$ such that for every $n \in [N]$

$$\left| \nabla_{q_{n,k}} L - \frac{M}{N} \widetilde{\nabla_{q_{n,k}} L}^{(n)} \right| \le \frac{4}{N} \sqrt{\frac{2\log(1/p_3) + 2\log(N) + 2\log(2)}{M}}.$$

This means that in the event $\mathcal{B}_k$, for every $m \in [M]$ we have

$$\left| \nabla_{q_{n_m,k}} L - \frac{M}{N} \nabla_{q_{n_m,k}} L^{(n)} \right| \le \frac{4}{N} \sqrt{\frac{2\log(1/p_3) + 2\log(N) + 2\log(2)}{M}}.$$

Lastly, given $0 < p < 1$, choosing $p_1 = p_3 = p/3K$ and $p_2 = p/3KD$ and intersecting all events for all corrdinates gives and event $\mathcal{E}$ of probability at least $1 - p$ such that

$$\left| \nabla_{r_k} L - \nabla_{r_k} L^{(n)} \right| \le 4 \sqrt{\frac{2\log(1/p_1) + 2\log(N) + 2\log(K) + 2\log(2)}{M}},$$

$$\left| \nabla_{f_{k,l}} L - \nabla_{f_{k,l}} L^{(n)} \right| \le \frac{4\lambda}{D} \sqrt{\frac{2\log(1/p_2) + 2\log(K) + 2\log(D) + 2\log(2)}{M}},$$

and

$$\left| \nabla_{q_{n_m,k}} L - \frac{M}{N} \nabla_{q_{n_m,k}} L^{(n)} \right| \le \frac{4}{N} \sqrt{\frac{2\log(1/p_3) + 2\log(N) + 2\log(K) + 2\log(2)}{M}}.$$

## F  Additional experiments

### F.1  Node classification

**Setup.** We evaluate ICG-NN and ICG$_\text{u}$-NN on the non-sparse graphs tolokers [43], squirrel [45] and twitch-gamers [36]. We follow the 10 data splits of Platonov et al. [43] for tolokers, the 10 data splits of Pei et al. [42], Li et al. [34] for squirrel and the 5 data splits of Lim et al. [36], Li et al. [34] for twitch-gamers. We report the mean ROC AUC and standard deviation for tolokers and the accuracy and standard deviation for squirrel and twitch-gamers. We also evaluate the relative Frobenius error between the graph and the ICG as a measure of the quality of the ICG approximation.

The baselines MLP, GCN [30], GAT [53], H$_2$GCN [69], GPR-GNN [14], LINKX [36], GloGNN [34] are taken from Platonov et al. [43] for tolokers and from Li et al. [34] for squirrel, twitch-gamers and penn94. We use the Adam optimizer and report all hyperparameters in Appendix I.

**Results.** All results are reported in Table 2. Observe that ICG-NNs achieve state-of-the-art results across the board, despite the low ratio of edge (graph densities of $2.41 \cdot 10^{-4}$ to $8.02 \cdot 10^{-3}$). ICG-NNs surpass the performance of more complex GNN models such as GT, LINKX and GloGNN, which solidify ICG-NNs as a strong method and a possible contender to message-passing neural networks.

### F.2  Comparison with graph coarsening methods over large graphs

In this section we highlight the main differences and similarities between graph coarsening methods and ICG-NNs, followed by an empirical comparison over large graphs.

#### F.2.1  Conceptual comparison

Graph coarsening methods [18, 25], graph condensation methods [26] and graph summarization methods [68] replace the original graph by one which has fewer nodes, with different features and a different topological structure, while sometimes preserving certain properties, such as the degree distribution [68].

**Theoretical guarantees.** Graph coarsening methods do not typically stem from theoretical guarantees, whereas ICG-NNs provably approximate the graph.

**Computational complexity.** Graph coarsening methods usually account for the graph's structure by apply message-passing on the coarsened graph. Condensation methods require $\mathcal{O}(EM)$ operations

Table 2: Results on large graphs. Top three models are colored by First, Second, Third.

| | tolokers | squirrel | twitch-gamers |
|---|---|---|---|
| # nodes | 11758 | 5201 | 168114 |
| # edges | 519000 | 217073 | 6797557 |
| avg. degree | 88.28 | 41.71 | 40.43 |
| relative Frob. | 0.69 | 0.39 | 0.8 |
| MLP | $72.95 \pm 1.06$ | $28.77 \pm 1.56$ | $60.92 \pm 0.07$ |
| GCN | $83.64 \pm 0.67$ | $53.43 \pm 2.01$ | $62.18 \pm 0.26$ |
| GAT | $83.70 \pm 0.47$ | $40.72 \pm 1.55$ | $59.89 \pm 4.12$ |
| GT | $83.23 \pm 0.64$ | - | - |
| $H_2$GCN | $73.35 \pm 1.01$ | $35.70 \pm 1.00$ | OOM |
| GPR-GNN | $72.94 \pm 0.97$ | $34.63 \pm 1.22$ | $61.89 \pm 0.29$ |
| LINKX | - | $61.81 \pm 1.80$ | $66.06 \pm 0.19$ |
| GloGNN | $73.39 \pm 1.17$ | $57.88 \pm 1.76$ | $66.34 \pm 0.29$ |
| ICG-NN | $83.73 \pm 0.78$ | $58.48 \pm 1.77$ | $65.27 \pm 0.82$ |
| ICG$_\text{u}$-NN | $83.51 \pm 0.52$ | $64.02 \pm 1.67$ | $66.08 \pm 0.74$ |

Table 3: Comparison with graph coarsening methods over large graphs. Top three models are colored by First, Second, Third.

| | Reddit | Flickr |
|---|---|---|
| # nodes | 232965 | 89250 |
| # edges | 11606919 | 899756 |
| avg. degree | 49.82 | 10.08 |
| Coarsening | $47.4 \pm 0.9$ | $44.6 \pm 0.1$ |
| Random | $66.3 \pm 1.9$ | $44.6 \pm 0.2$ |
| Herding | $71.0 \pm 1.6$ | $44.4 \pm 0.6$ |
| K-Center | $58.5 \pm 2.1$ | $44.1 \pm 0.4$ |
| One-Step | - | $45.4 \pm 0.3$ |
| DC-Graph | $90.5 \pm 1.2$ | $45.8 \pm 0.1$ |
| GCOND | $90.1 \pm 0.5$ | $47.1 \pm 0.1$ |
| SFGC | $89.9 \pm 0.4$ | $47.1 \pm 0.1$ |
| GC-SNTK | - | $46.6 \pm 0.2$ |
| ICG-NN | $89.6 \pm 1.2$ | $50.4 \pm 0.1$ |
| ICG$_\text{u}$-NN | $93.6 \pm 1.2$ | $52.7 \pm 0.1$ |

to construct a smaller graph [26, 67, 56], where $E$ is the number of edges of the original graph and $M$ is the number of nodes of the condensed graph. Conversely, ICGs estimate the ICG with $\mathcal{O}(K^2N + KE)$ operations, where $K$ is typically smaller than $M$.

**Node processing.** Graph coarsening methods process representations on an iteratively coarsened graph, while ICG-NN also process the fine node information at every layer.

**Graphs larger than memory.** ICG-NNs offer a subgraph sampling approach when the original graph cannot fit in memory. In contrast, the aforementioned methods lack a strategy for managing smaller data structures when computing the compressed graph.

### F.2.2 Empirical comparison

**Setup.** We evaluate ICG-NN and ICG$_\text{u}$-NN on the large graph datasets Flickr [65] and Reddit [23]. We follow the splits of [67] and report the accuracy and standard deviation.

The standard graph coarsening baselines Coarsening [25], Random [25], Herding [58], K-Center [48], One-Step [27] and the graph condensation baselines DC-Graph [66], GCOND [26], SFGC [67] and GC-SNTK [56] are taken from [67, 56]. We use the Adam optimizaer and report all hyperparameters in Appendix I.

**Results.** ICG-NNs present state-of-the-art performance in Table 3 when compared to a vareity of both graph coarsening methods and graph condensation methods, further solidifying its effectiveness.

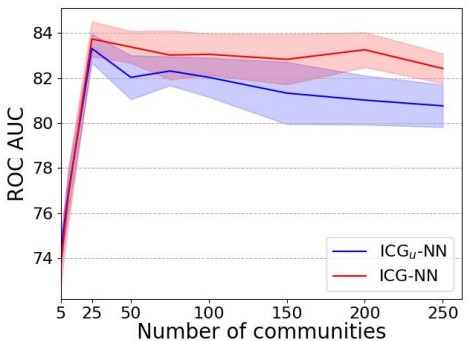 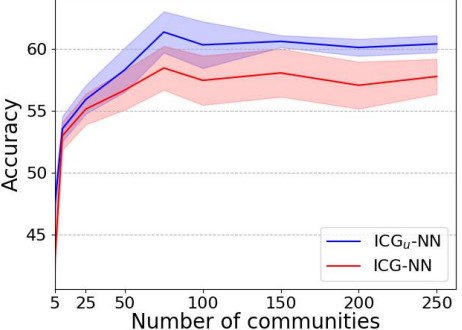

Figure 4: Test ROC AUC of tolokers (**left**) and test accuracy of squirrel (**right**) as a function of the number of communities.

### F.3 Node classification on large graphs using Subgraph SGD

**Setup.** We evaluate ICG-NN and ICG$_{\mathrm{u}}$-NN on the large graph Flickr [65]. We follow the splits of [67] and report the accuracy and standard deviation. We set the ratio of condensation $r = \frac{M}{N}$ to 1%, where $N$ and $M$ are the number of nodes in the graph and the number of sampled nodes.

The graph coarsening baselines Coarsening [25], Random [25], Herding [58], K-Center [48], One-Step [27] and the graph condensation baselines DC-Graph [66], GCOND [26], SFGC [67] and GC-SNTK [56] are taken from [67, 56]. We use the Adam optimizaer and report all hyperparameters in Appendix I.

**Results.** Table 4 shows that ICG$_{\mathrm{u}}$-NN with subgraph ICGm with 1% sampling rate, achieves competitive performance with coarsening and condensation methods that operate on the full graph in memory.

Table 4: Results on Flickr using 1% node sampling. Top three models are colored by First, Second, Third.

| Model | Accuracy |
|---|---|
| Coarsening | $44.6 \pm 0.1$ |
| Random | $44.6 \pm 0.2$ |
| Herding | $44.4 \pm 0.6$ |
| K-Center | $44.1 \pm 0.4$ |
| One-Step | $45.4 \pm 0.3$ |
| DC-Graph | $45.8 \pm 0.1$ |
| GCOND | $47.1 \pm 0.1$ |
| SFGC | $47.1 \pm 0.1$ |
| GC-SNTK | $46.5 \pm 0.2$ |
| ICG-NN | $50.4 \pm 0.1$ |
| ICG$_{\mathrm{u}}$-NN | $50.8 \pm 0.1$ |

### F.4 Ablation study over the number of communities

**Setup.** We evaluate ICG-NN and ICG$_{\mathrm{u}}$-NN on non-sparse graphs tolokers [43] and squirrel [45]. We follow the 10 data splits of Platonov et al. [43] for tolokers and Pei et al. [42], Li et al. [34] for squirrel. We report the the mean ROC AUC and standard deviation for tolokers and the accuracy and standard deviation for squirrel.

**Results.** Figure 4 shows several key trends across both datasets. First, as anticipated, when a very small number of communities is used, the ICG fails to approximate the graph accurately, resulting in degraded performance. Second, the optimal performance is observed when using a relatively small number of communities, specifically 25 for tolokers and 75 for squirrel. Lastly, using a large number of communities does not appear to negatively impact performance.

### F.5 The cut-norm and Frobenius norm

While the cut norm is the theoretical target of the ICG approximation, it is expensive to estimate directly. However, Theorem 3.1 implies that optimizing the Frobenius norm, which is computationally inexpensive, leads to a small cut norm.

In this experiment, we first aim to demonstrate the positive correlation between the Frobenius error and the cut norm error. Moreover, we show that even though the Frobenius error cannot be made close to zero in general, it is still correlated with cut norm. We show that when the relative Frobenius

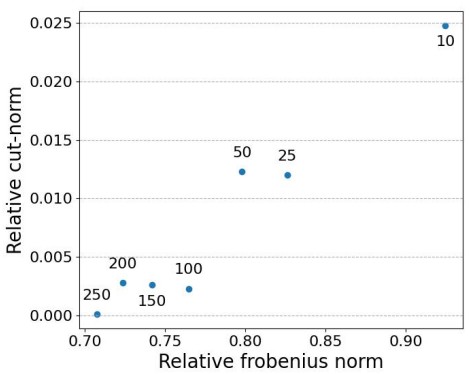 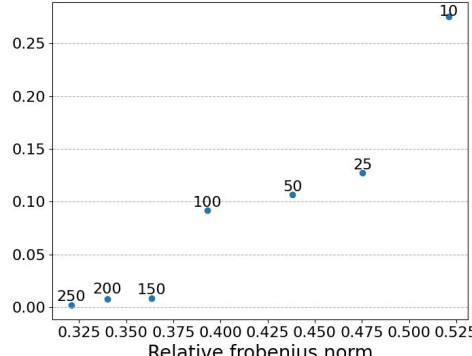

Figure 5: Cut-norm as a function of Frobenius norm on the tolokers (**left**) and squirrel (**right**) datasets. The number of communities used is indicated next to each point.

Table 5: Time until convergence in seconds on large graphs.

|  | tolokers | squirrel | twitch-gamers |
|---|---|---|---|
| # nodes | 11758 | 5201 | 168114 |
| # edges | 519000 | 217073 | 6797557 |
| avg. degree | 88.28 | 41.71 | 40.43 |
| random init. | $83.30 \pm 0.92$ | $62.03 \pm 0.98$ | $64.73 \pm 0.44$ |
| eigenvector init. | $83.31 \pm 0.64$ | $62.10 \pm 1.67$ | $66.10 \pm 0.42$ |

error is reduced below a specified threshold (not close to zero in general), the ICG approximation allows ICG-NNs to achieve state-of-the-art results.

**Setup.** We evaluate our ICG approximation on non-sparse graphs tolokers [43] and squirrel [45]. We vary the number of communities used by the approximation and report the relative cut-norm (estimated by [3]) as a function of the relative Frobenius norm. We also present the resulting relative Frobenius error for our node classification experiments in Table 2.

**Results.** As expected, the cut-norm and Frobenius norm are positively correlated, both decreasing as more communities are used, as shown in Figure 5. We also observe in Table 2 that ICG-NNs perform best on the squirrel dataset and worst on twitch-gamers, aligning with their relative Frobenius error. The lower the relative Frobenius error, the more accurate the ICG approximation, leading to better performance. Table 2 and Figure 5 also revealed a useful rule of thumb: when the relative Frobenius error is below 0.8, it correlates with a very small cut norm error. This, in turn, leads to an accurate ICG approximation, enabling ICG-NNs to achieve state-of-the-art results.

### F.6 Initialization

We repeated the node classification experiments presented in Table 2, using random initialization when optimizing the ICG, and compared the results to those obtained with eigenvector initialization.

**Setup.** We evaluate ICG-NN and $ICG_u$-NN on the non-sparse graphs tolokers [43], squirrel [45] and twitch-gamers [36]. We follow the 10 data splits of Platonov et al. [43] for tolokers, the 10 data splits of Pei et al. [42], Li et al. [34] for squirrel and the 5 data splits of Lim et al. [36], Li et al. [34] for twitch-gamers. We report the mean ROC AUC and standard deviation for tolokers and the accuracy and standard deviation for squirrel and twitch-gamers.

**Results.** Table 5 indicates that eigenvector initialization generally outperforms random initialization across all datasets. While the improvements are minimal in some cases, such as with the tolokers and squirrel datasets, they are more pronounced in the twitch-gamers dataset.

Additionally, eigenvector initialization offers a practical advantage in terms of training efficiency. On average, achieving the same loss value requires 5% more time when using random initialization

compared to eigenvector initialization. This efficiency gain further supports the utility of the proposed initialization method.

## F.7  Run-time analysis

In this subsection, we compare the forward pass run times of our ICG approximation process, signal processing pipeline (ICG$_\text{u}$-NN) and the GCN [30] architecture. We sample graphs of sizes up to 7,000 from a dense and a sparse *Erdős-Rényi* distribution $\text{ER}(n, p(n) = 0.5)$ and $\text{ER}(n, p(n) = \frac{50}{n})$, correspondingly. Node features are independently drawn from $U[0, 1]$ and the initial feature dimension is 128. ICG$_\text{u}$-NN and GCN use a hidden dimension of 128, 3 layers and an output dimension of 5.

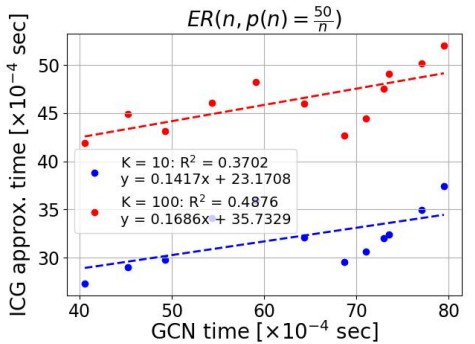
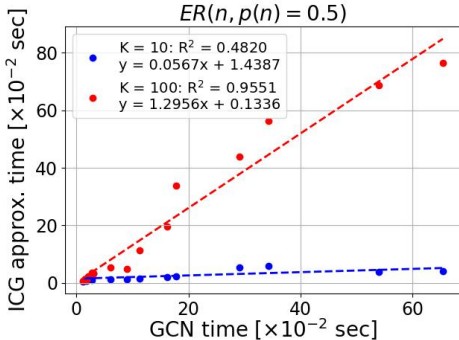

Figure 6: Empirical runtimes: K-ICG approximation process as a function of GCN forward pass duration on the dense (**left**) and sparse (**right**) *Erdős-Rényi* distribution $\text{ER}(n, p(n) = 0.5)$ and $\text{ER}(n, p(n) = \frac{50}{n})$ for K=10, 100.

**Results.** Figure 6 demonstrates a strong linear relationship between the runtime of our ICG approximation and that of the message-passing neural network GCN for both sparse and dense graphs. Unlike GCN, our ICG approximation is versatile for multiple tasks and requires minimal hyperparameter tuning, which reduces its overall time complexity. Additionally, Appendix F.7 reveals a strong square root relationship between the runtime of ICG$_\text{u}$-NN and the runtime of GCN for both sparse and dense graphs. This aligns with our expectations, as the time complexity of GCN is $\mathcal{O}(E)$, while the time complexity of ICG-NNs is $\mathcal{O}(N)$, highlighting the computational advantage of using ICGs over message-passing neural networks.

## F.8  Time until convergence

**Setup.** We measured the average time until convergence in seconds for the GCN and ICG$_\text{u}$-NN architectures on the non-sparse graphs tolokers [43], squirrel [45] and twitch-gamers [36]. We follow

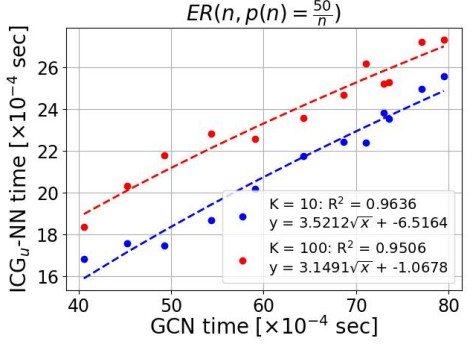
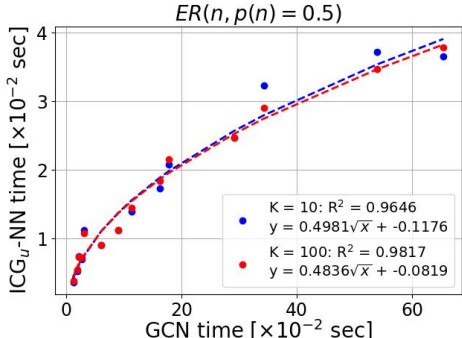

Figure 7: Empirical runtimes: K-ICG$_\text{u}$-NN as a function of GCN forward pass duration on the dense (**left**) and sparse (**right**) *Erdős-Rényi* distribution $\text{ER}(n, p(n) = 0.5)$ and $\text{ER}(n, p(n) = \frac{50}{n})$ for K=10, 100.

Table 6: Time until convergence in seconds on large graphs.

|  | tolokers | squirrel | twitch-gamers |
|---|---|---|---|
| GCN | 510 | 3790 | 1220 |
| $ICG_u$-NN | 33 | 225 | 49 |

the 10 data splits of Platonov et al. [43] for tolokers, the 10 data splits of Pei et al. [42], Li et al. [34] for squirrel and the 5 data splits of Lim et al. [36], Li et al. [34] for twitch-gamers. We ended the training after 50 epochs in which the validation metric did not improve. We set the hyper-parameters with best validation results.

**Results.** Table 6 shows that $ICG_u$-NN consistently converges faster than GCN across all three benchmarks. Specifically, converges approximately 15 times faster on the tolokers dataset, 17 times faster on the squirrel dataset, and 25 times faster on the twitch-gamers dataset compared to GCN, indicating a significant improvement in efficiency.

### F.9 Memory allocation analysis

In this subsection, we compare the memory allocated during the ICG approximation process and during a forward pass of the GCN [30] architecture. We sample graphs of sizes up to 2,000 from a dense *Erdős-Rényi* distribution $ER(n, p(n) = 0.5)$. Node features are independently drawn from $U[0, 1]$ and the initial feature dimension is 128. $ICG_u$-NN and GCN use a hidden dimension of 128, 3 layers and an output dimension of 5.

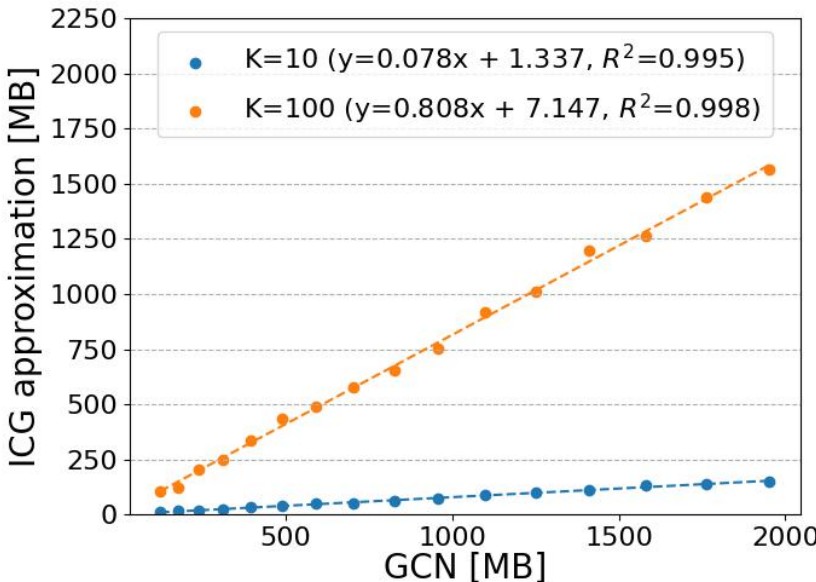

Figure 8: Memory allocated for K-ICG approximation (K=10,100) as a function of the memory allocated for GCN on graphs $G \sim ER(n, p(n) = 0.5)$.

**Results.** Appendix F.9 demonstrates a strong linear relationship between the the memory allocated during the ICG approximation process and during a forward pass of the message-passing neural network GCN. This aligns with our expectations: the memory allocation for the GCN architecture is $\mathcal{O}(ED)$, and of ICG-NNs $\mathcal{O}(EK + NKD)$, where $D$ is the feature dimension and $K$ is the number of communities used for the ICG approximation.

# G  Additional related work

**Latent graph GNNs.** One way to interpret our ICG-NNs are as latent graph methods, where the ICG approximation of the given graph $G$ is seen as a latent low complexity "clean version" of the observed graph $G$.

In latent graph methods, the graph on which the GNN is applied is inferred from the given data graph, or from some raw non-graph data. Two motivations for latent graph methods are (1) the given graph is noisy, and the latent graph is its denoised version, or (2) even if the given graph is "accurate", the inferred graph improves the efficiency of message passing. Our ICG GNN can be cast in category (2), since we use the latent ICG model to define an efficient "message passing" scheme, which has linear run-time with respect to the number of nodes.

Some examples of latent graph methods are papers like [31, 57, 16, 10, 17] which learn a latent graph from non-graph raw data. Papers like [39, 59, 28, 21] treat the given graph as noisy or partially observed, and learn the "clean" graph. Papers like [54, 50] modify the given graph to improve the efficiency of message passing (e.g., based on the notion of oversquashing [5]), and [41] modifies the weights of the given graph if the nodes were sampled non-uniformly from some latent geometric graph. Additionaly, attention based GNN methods, which dynamically choose edge weights [53, 55, 52, 64], can be interpreted as latent graph approaches.

# H  Dataset statistics

The statistics of the real-world node-classification, spatio-temporal and graph coarsening benchmarks used can be found in Tables 7 to 9.

Table 7: Statistics of the node classification benchmarks.

|  | tolokers | squirrel | twitch-gamers |
|---|---|---|---|
| # nodes (N) | 11758 | 5201 | 168114 |
| # edges (E) | 519000 | 217073 | 6797557 |
| avg. degree ($\frac{E}{N}$) | 88.28 | 41.71 | 40.32 |
| # node features | 10 | 2089 | 7 |
| # classes | 2 | 5 | 2 |
| metrics | AUC-ROC | ACC | ACC |

Table 8: Statistics of the spatio-temporal benchmarks.

|  | METR-LA | PEMS-BAY |
|---|---|---|
| # nodes (N) | 207 | 325 |
| # edges (E) | 1515 | 2369 |
| avg. degree ($\frac{E}{N}$) | 7.32 | 7.29 |
| # node features | 34272 | 52128 |

Table 9: Statistics for graph coarsening benchmarks.

|  | Reddit | Flickr |
|---|---|---|
| # nodes (N) | 232965 | 89250 |
| # edges (E) | 114615892 | 899756 |
| avg. degree | 491.99 | 10.08 |
| # node features | 602 | 500 |
| # classes | 41 | 7 |

# I Hyperparameters

In Tables 10 to 12, we report the hyper-parameters used in our real-world node-classification, spatio-temporal and graph coarsening benchmarks.

Table 10: Hyper-parameters used for the node classification benchmarks.

|  | tolokers | squirrel | twitch-gamers |
|---|---|---|---|
| # communities | 25 | 75 | 50 |
| Encoded dim | - | 128 | - |
| $\lambda$ | 1 | 1 | 1 |
| Approx. lr | 0.01, 0.05 | 0.01, 0.05 | 0.01, 0.05 |
| Approx. epochs | 10000 | 10000 | 10000 |
| # layers | 3,4,5 | 3,4,5 | 3,4,5,6 |
| hidden dim | 32, 64 | 64, 128 | 64, 128 |
| dropout | 0, 0.2 | 0, 0.2 | 0, 0.2 |
| residual connection | - | ✓ | ✓ |
| Fit lr | 0.001, 0.003, 0.005 | 0.001, 0.003, 0.005 | 0.001, 0.003, 0.005 |
| Fit epochs | 3000 | 3000 | 3000 |

Table 11: Hyper-parameters used for the spatio-temporal benchmarks.

|  | METR-LA | PEMS-BAY |
|---|---|---|
| # communities | 50 | 100 |
| Encoded dim | - | - |
| $\lambda$ | 0 | 0 |
| Approx. lr | 0.01, 0.05, 0.1 | 0.01, 0.05, 0.1 |
| Approx. epochs | 10000 | 10000 |
| # layers | 3,4,5 | 3,4,5, 6 |
| hidden dim | 32, 64, 128 | 32, 64, 128 |
| Fit lr | 0.001, 0.003 | 0.001, 0.003 |
| Fit epochs | 300 | 300 |

Table 12: Hyper-parameters used for graph coarsening benchmarks.

|  | Reddit | Flickr |
|---|---|---|
| # communities | 50, 600 | 50, 500 |
| Encoded dim | - | - |
| $\lambda$ | 0 | 0,1 |
| Approx. lr | 0.05 | 0.01, 0.05 |
| Approx. epochs | 10000 | 10000 |
| # layers | 2, 3 | 2, 3, 4 |
| hidden dim | 128, 256 | 128, 256 |
| dropout | 0, 0.2 | 0 |
| residual connection | ✓ | ✓ |
| Fit lr | 0.005, 0.01 | 0.001, 0.003, 0.005 |
| Fit epochs | 3000 | 1500 |

For spatio-temporal dataset, we follow the methodology described in [15], where the time of the day and the one-hot encoding of the day of the week are used as additional features.

