# OpenReview forum: "Learning on Large Graphs using Intersecting Communities"
_NeurIPS.cc/2024/Conference — NeurIPS 2024 poster_

### Official Review · Reviewer_XnC3 · 2024-07-09

**Soundness:** 3
**Presentation:** 3
**Contribution:** 2
**Rating:** 5
**Confidence:** 4

**Summary:**

This paper introduces a novel approach for graph machine learning on large graphs using a concept called Intersecting Community Graphs (ICG). Traditional MPNNs face memory and computational challenges when dealing with large graphs due to their reliance on graph edges. The proposed method approximates the input graph as a combination of ICG, which requires fewer communities than the size of the graph. The ICG approach leverages a new constructive version of the Weak Graph Regularity Lemma to construct these approximations efficiently. The resultant learning algorithm operates with memory and time complexity linear to the number of nodes, rather than edges, making it suitable for very large, non-sparse graphs. Empirical evaluations demonstrate the method's applicability to tasks like node classification and spatio-temporal data processing.

**Strengths:**

1. In theory, the approach scales linearly with the number of nodes, addressing the memory complexity issues that plague traditional MPNNs when applied to large graphs.

2. The paper introduces a constructive version of the Weak Graph Regularity Lemma, which could be of independent interest in graph theory and its applications.

3. The method has been empirically validated on real-world tasks, showing competitive performance against existing models in node classification and spatio-temporal graph tasks.

**Weaknesses:**

1. The proposed method's speed advantage over MPNNs hinges on the assumption that the chosen number of communities K should be less than the node degree (line 170). However, real-world experiments sometimes require a K value larger than the node degree. For example, in the twitch-gamers dataset, K is set to 500 while the average node degree is 40.32. This discrepancy raises concerns about the practical speed benefits of the proposed method on real-world benchmarks.

2. The computation of the ICG requires an offline pre-computation step that, while only performed once, involves a time complexity linear to the number of edges.

3. In the Runtime Analysis section, the authors should also include a time comparison for the model's convergence, in addition to the per-epoch runtime.


4. To demonstrate the efficacy of the proposed initialization method, the paper should include a comparison with existing initialization methods, such as random initialization.

**Questions:**

Has the author evaluated the relationship between the model's performance and the cut-metric between the approximated graph and the original graph? While I agree that in certain situations, such as social networks, ICG can approximate the original graph well, it is important to question whether this assumption holds in other contexts, such as molecular graphs.

**Limitations:**

N/A.

---

> ### Author Rebuttal · Authors · 2024-08-06
>
> We thank the reviewer for their comments and for acknowledging the theoretical significance of our constructive version of the Weak Graph Regularity Lemma. We address each of their questions/concerns below.
>
> > W1: The proposed method's speed advantage over MPNNs hinges on the assumption that the chosen number of communities K should be less than the node degree (line 170)...
>
> **Answer:** Thanks for raising this point, this was an oversight. We conducted further experiments with a significantly lower number of communities $K=50$, achieving an accuracy of 66.08 ± 0.74 and 65.27 ± 0.82 for $\text{ICG}_u-\text{NN}$ and ICG-NN, correspondingly. Thus achieving state-of-the-art results while retaining the speed benefits of ICG-NN.
>
> > W2: The computation of the ICG requires an offline pre-computation step that, while only performed once, involves a time complexity linear to the number of edges.
>
> **Answer:** That is accurate. The time complexity of the ICG fitting process is similar to the time complexity of MPNNs, linear in the number of edges. Let us clarify that this is a strength of ICG-NNs. The ICG fitting process is only done once and does not require extensive hyperparameter optimization and architecture tuning, unlike MPNNS. Thus, ICG-NN offer a potentially significant advantage over standard MPNNs. This gap between ICG-NNs and MPNNs is further amplified for time series on graphs.
>
> > W3: In the Runtime Analysis section, the authors should also include a time comparison for the model's convergence, in addition to the per-epoch runtime.
>
> **Answer:** Based on the reviewers' suggestion, we measured the time (in seconds) until convergence for the GCN and $\text{ICG}_u-\text{NN}$ architectures across the node-classification tasks presented in **Appendix F.1, Table 2**. We ended the training after 50 epochs in which the validation metric didn’t improve. We set the hyper-parameters with best validation results. The results are shown in the following table:
>
> |       | tolokers | squirrel | twitch-gamers |
> |-------|----------|----------|---------------|
> | GCN   | 510      | 3790     | 1220          |
> | ICGNN | 33       | 225      | 49            |
>
> It is clear that $\text{ICG}_u-\text{NN}$ consistently converges faster than GCN across all three benchmarks. Specifically, $\text{ICG}_u-\text{NN}$ converges approximately 15 times faster on the **tolokers** dataset, 17 times faster on the **squirrel** dataset, and 25 times faster on the **twitch-gamers** dataset compared to GCN, indicating a significant improvement in efficiency.
> We will add this to the revised paper.
>
> > W4: To demonstrate the efficacy of the proposed initialization method, the paper should include a comparison with existing initialization methods, such as random initialization.
>
> **Answer:** We repeated the node classification experiments presented in **Table 2**, as described in **Appendix F.1**, using random initialization when optimizing the ICG, and compared the results to those obtained with eigenvector initialization.
>
> |                            | tolokers     | squirrel     | twitch-gamers |
> |----------------------------|--------------|--------------|---------------|
> | random initialization      | 83.30 ± 0.92 | 62.03 ± 0.98 | 64.73 ± 0.44  |
> | eigenvector initialization | 83.31 ± 0.64 | 62.10 ± 1.67 | 66.10 ± 0.42  |
>
> The results indicate that eigenvector initialization generally outperforms random initialization across all datasets. While the improvements are minimal in some cases, such as with the tolokers and **squirrel** datasets, they are more pronounced in the **twitch-gamers** dataset.
>
> Additionally, eigenvector initialization offers a practical advantage in terms of training efficiency. On average, achieving the same loss value requires 5% more time when using random initialization compared to eigenvector initialization. This efficiency gain further supports the utility of the proposed initialization method.
>
> We will add these results and discussion to an ablation study section in the appendix.
>
> > Q1: Has the author evaluated the relationship between the model's performance and the cut-metric...
>
> **Answer:** **Figure 4** in **Appendix F.2** shows the relationship between test accuracy and the number of communities used, while **Figure 5** in **Appendix F.3** illustrates the relationship between cut-norm, Frobenius norm, and the number of communities. Using these figures, we present the accuracy as a function of the cut-norm, as requested by the reviewer:
>
> | cut-norm error | 0.09 | 0.1  | 0.11 | 0.13 | 0.27 |
> |----------|------|------|------|------|------|
> | accuracy | 61.3 | 62.1 | 57.2 | 55.8 | 53.2 |
>
> We observe that as the cut-norm error increases, the test accuracy generally decreases. This indicates that a lower cut-norm, which corresponds to a more refined community structure, tends to result in higher accuracy of the model, as expected.
>
> > it is important to question whether this assumption holds in ther contexts, such as molecular graphs
>
> The setting in this paper is a single large graph. Our setting does not extend to a dataset of many graphs. Molecular graphs are typically used in graph classification or regression, which is not appropriate for our method. This is emphasised in Line 44.

---

> > ### Comment · Reviewer_XnC3 · 2024-08-10
> >
> > I thank the authors for their rebuttal, which partially addresses my concerns. In general, the paper contributes insights to the field, but there are still areas for improvement, such as further validation to ensure the robustness and scalability of the model. As a result, I will maintain my current score.

---

> > > ### Author Response · Authors · 2024-08-11
> > >
> > > > In general, the paper contributes insights to the field
> > >
> > > We thank the reviewer for acknowledging the contribution of this work to the field.
> > >
> > > > there are still areas for improvement, such as further validation to ensure the robustness and scalability of the model.
> > >
> > > To further validate and ensure the robustness and scalability of our model, we conducted additional experiments which were not previously mentioned in our rebuttal to your review, testing ICG-NNs and their subgraph variation:
> > > * **Comparison to graph coarsening methods** - We provide an empirical comparison in **Table 1** of **rebuttal.pdf** between our method and a variety of graph coarsening/summarization methods on the **reddit** and **flickr** datasets. ICG-NNs achieve state-of-the-art performance, further solidifying its effectiveness.
> > > * **Subgraph ICG-NN and ICG-NN on the large graph flickr** - To highlight the usage of subgraph sgd ICG-NN on large graphs, we conducted additional experiments using the **flickr** dataset, which is significantly larger than the datasets used in our initial submission. The results indicate that ICG-NNs with subgraph ICGm, using a 1% sampling rate, outperform competing methods that operate on the full graph.
> > >
> > > > I thank the authors for their rebuttal, which partially addresses my concerns.
> > >
> > > We would like to use this opportunity to clarify remaining concerns. Could you give us specific issues you are concerned with, so we have the opportunity to address them?
> > >
> > >
> > > In light of this we would greatly appreciate a reconsideration of the current score.

---

### Official Review · Reviewer_hec5 · 2024-07-10

**Soundness:** 2
**Presentation:** 3
**Contribution:** 3
**Rating:** 6
**Confidence:** 4

**Summary:**

The paper proposes a novel method to improve the GNN efficiency for large and dense graphs by approximating them as Intersecting Community Graphs (ICGs). This approach significantly reduces time and memory complexity depending on the number of nodes rather than edges. The authors theoretically show the approximation of ICGs using the weak graph regularity lemma. Experiments demonstrate ICG’s efficiency in node classification and spatio-temporal data tasks.

**Strengths:**

The authors propose an efficient method to train large (and dense) graphs. The method is theoretically well-motivated and concretely proved with detailed explanations. The theory is interesting, and the techniques used can be applied to future work in various ways. In addition, there are diverse types of benchmarks and evaluations (runtime analysis, node classification, spatio-temporal graphs), and the paper presents their results clearly.

**Weaknesses:**

- From my perspective, ICG+GNN is similar to summarizing the graph into smaller units (e.g., coarsened nodes, communities, or subgraphs) and then performing message-passing between them. It is essential to acknowledge the line of research on GNNs with graph coarsening, summarization, and condensation. These methods have addressed the challenges in graph learning, especially for efficiency and scalability. The examples of recent papers are stated below, and some of the work should be compared with empirical experiments.
  - Hierarchical Inter-Message Passing for Learning on Molecular Graphs (ICML workshop 2020)
  - Scaling Up Graph Neural Networks Via Graph Coarsening (KDD 2021)
  - GRAPH CONDENSATION FOR GRAPH NEURAL NETWORKS (ICLR 2022)
  - A Provable Framework of Learning Graph Embeddings via Summarization (AAAI 2023)
  - Gapformer: Graph transformer with graph pooling for node classification (IJCAI 2023)
  - Structure-free Graph Condensation: From Large-scale Graphs to Condensed Graph-free Data (NeurIPS 2023)
  - Fast Graph Condensation with Structure-based Neural Tangent Kernel (WWW 2024)
  - Translating Subgraphs to Nodes Makes Simple GNNs Strong and Efficient for Subgraph Representation Learning (ICML 2024)
- The time and memory required for real runtime to fit ICGs are not demonstrated. It makes sense that we only need to create the ICG once, but how much time and GPU memory does it take to create it?
- Evaluation across datasets should be encouraged. Although various evaluations have been performed, a set of datasets was picked for each evaluation. A natural question arises as to whether this method robustly works across datasets.
- Scalability should be validated by experiments on larger datasets and be compared with other efficient GNNs. The authors stated that constraints (single, large, undirected) restrict the number of datasets the authors can experiment on. But as far as I know, there are many more single, large, undirected graph datasets. Is the largest dataset used in the experiment large enough compared to existing efficient GNN research? Plus, is this method superior to existing scalable GNNs (e.g., sampling, coarsening) in practice (i.e., total training time)?

**Questions:**

- p249: Why does the operation on Community feature vector F is O(K^2D^2) rather than O(KD^2)? F is a matrix, the shape of which is K x D, then MLP on this matrix will be computed in O(KD^2) like p252.
- p253: Why is Q^{†}Q an identity matrix? Are the rows of Q linearly independent? I could not find the constraint or assumption on this.
- Notations: How about using more distinguishable notations for node features S and signals \italic{S}? This might not be clear to readers.

**Limitations:**

The authors adequately addressed the limitations.

---

> ### Author Rebuttal · Authors · 2024-08-06
>
> We thank the reviewer for their comments and for finding our work theoretically significant.
>
> > W1: ICG+GNN is similar to summarizin…
>
> **Answer:** In **Comparison of ICG-NN to graph coarsening** in the common response, we highlight the main differences to pooling methods.
>
> > … The examples of recent papers are stated below…
>
> **Answer:** We agree, and for that reason we mentioned intersecting communities, stochastic block models and GNNs with local pooling in our related work sections: **Section 7** and **Appendix I**.
>
> Let us highlight the fundamental differences between the mentioned papers and ours:
> 1. **Computational complexity** - Condensation methods require $O(EM)$ operations to construct a smaller graph [3,6,7,8], where $E$ is the number of edges of the original graph and $M$ is the number of nodes of the condensed graph. Conversely, we estimate the ICG with $O(E)$ operations.
> 2. **Graph construction** - Methods like coarsening [1,2], condensation [3] and summarization [4], typically rely on heuristics. In contrast, ICGs provide a provable approximation of the original graph.
> 3. **Handling graphs that do not fit in memory** - ICG-NNs offer a subgraph sampling approach when the original graph cannot fit in memory. In contrast, the aforementioned methods lack a strategy for managing smaller data structures when computing the compressed graph.
>
> In line with the reviewer's suggestion, we will include the proposed references with an appropriate discussion.
>
> In addition, we provide an empirical comparison in **Table 1** of **rebuttal.pdf** between our method and the aforementioned approaches on the **reddit** and **flickr** datasets. ICG-NNs achieve state-of-the-art performance.
>
> > W2: The time and memory required for real runtime to fit ICGs…
>
> **Answer:** In Appendix F.4 we compare the forward pass run times of our ICG approximation process, signal processing pipeline ($\text{ICG}_u-\text{NN}$) and the GCN architecture, demonstrating a linear relationship between the two.
>
> Following your request, we estimated the GPU memory used while fitting an ICG. We follow a similar setup to the run time experiments, with Erdos-Rényi graphs with 128-dimensional node features sampled from $U[0,1]$. Both $\text{ICG}_u-\text{NN}$ and GCN use a hidden dimension of 128, 3 layers and an output dimension of 5:
>
> **Figure 1** in **rebuttal.pdf** reveals a linear relationship between the memory allocated for $\text{ICG}_u-\text{NN}$ and for GCN. This aligns with our expectations: the memory complexity of GCN is $O(Ed)$, and of ICG-NNs $O(EK)$.
>
> > W3: Evaluation across datasets...
>
> > W4: …there are many more single, large, undirected graph...
>
> > Is the largest dataset used in the experiment large enough…
>
> **Answer:** We conducted additional experiments using the **reddit** and **flickr** datasets, which are significantly larger than the datasets used in our initial submission. We provide an empirical comparison in **Table 1** of **rebuttal.pdf** between our method and existing efficient approaches. ICG-NNs achieve state-of-the-art performance, further solidifying its effectiveness.
>
> Our method is designed for large and non-sparse graphs. Unfortunately, very few datasets meet this criterion. Standard benchmarks of large graphs are quite sparse. However, in applications in the industry, one often has to work with large and not very sparse graphs. Remarkably, ICG-NN still performs very well on the available sparse large graphs. In the paragraph in line 173, we explain how ICGs are still often meaningful for sparse graphs in practice. See **Choice of datasets** in the common response for more details.
>
> > Plus, is this method superior to existing scalable GNNs…
>
> **Answer:** We measured the time until convergence for the GCN and $\text{ICG}_u-\text{NN}$ architectures and tasks presented in **Appendix F.1, Table 2**. We ended the training after 50 epochs in which the validation metric didn’t improve. We set the hyper-parameters with best validation results. The results:
>
> |       | tolokers | squirrel | twitch-gamers |
> |-------|----------|----------|---------------|
> | GCN   | 510      | 3790     | 1220          |
> | ICGNN | 33       | 225      | 49            |
>
> $\text{ICG}_u-\text{NN}$ consistently converges faster than GCN across all three benchmarks. We will add this to the revised paper.
>
> > Q1: p 249: Why does the operation on Community feature vector $F$ is $O\left(K^2D^2\right)$...
>
> **Answer:**  When defining the general MLP, $F$ is flattened into a $KD$ dimensional vector. A linear operator hence requires $KD \times KD$ parameters. We will clarify this in the text.
>
> > Q2: $p$ 253: Why is $Q^\dagger Q$ an identity…
>
> **Answer:** If $Q$ is full rank at initialization (which is true almost surely for random initialization), and if $A$ is not low rank, then the optimal configuration of $Q$ is full rank. The optimization would not benefit from having multiple instances of the same community. This would be equivalent to reducing the number of communities from $K$ to $K-1$. We will clarify this in the paper.
>
> > Q3: How about using more distinguishable notations...
>
> **Answer:** We follow the standard numerical linear algebra notation, where $\mathbf{s}_i$ denotes the i-th row/column of the matrix $\mathbf{S}$ and it is standard to use the same letter for both.
>
> [1] Hierarchical Inter-Message Passing for Learning on Molecular Graphs, 2020.\
> [2] Scaling Up Graph Neural Networks Via Graph Coarsening, 2021.\
> [3] Graph Condensation for Graph Neural Networks, 2022.\
> [4] A Provable Framework of Learning Graph Embeddings via Summarization, 2023.\
> [5] Gapformer: Graph Transformer with Graph Pooling for Node Classifcation, 2023.\
> [6] Structure-free Graph Condensation: From Large-scale Graphs to Condensed Graph-free Data, 2023.\
> [7] Fast Graph Condensation with Structure-based Neural Tangent Kernel, 2024.\
> [8] Translating Subgraphs to Nodes Makes Simple GNNs Strong and Efficient for Subgraph Representation Learning, 2024.

---

> > ### Comment · Reviewer_hec5 · 2024-08-12
> >
> > I acknowledged the author's response. Thank you for the detailed explanations and they resolved my concerns. I will raise my score.
> >
> > Please include our discussions on your camera-ready paper. Looking forward to seeing the revised paper!

---

> > > ### Author Response · Authors · 2024-08-12
> > >
> > > > I acknowledged the author's response. Thank you for the detailed explanations and they resolved my concerns. I will raise my score.
> > >
> > > We warmly thank the reviewer for acknowledging our rebuttal and raising their score.
> > >
> > > > Please include our discussions on your camera-ready paper. Looking forward to seeing the revised paper!
> > >
> > > We will make sure to include all of the valuable points that the reviewers mentioned during the rebuttal in the final version of our paper.
> > >
> > > We would like to thank the reviewer again.

---

> ### Author Response · Authors · 2024-08-11
>
> We thank the reviewer for their comments and for acknowledging our work’s theoretical novelty and its future work potential. Since the author-reviewer discussion period is closing soon, we would highly appreciate feedback on our response. We are keen to take advantage of the author-reviewer discussion period to clarify any additional concerns.

---

### Official Review · Reviewer_2roR · 2024-07-12

**Soundness:** 3
**Presentation:** 3
**Contribution:** 4
**Rating:** 5
**Confidence:** 3

**Summary:**

This paper proposes the Intersecting Communities Graph (ICG), which enables efficient learning on very large non-sparse graphs by representing the graph as a linear combination of intersecting communities such as cliques. Unlike Message Passing Neural Networks (MPNNs), the proposed method operates with memory complexity proportional to the number of nodes, even for extremely large graphs. Although constructing the ICG takes linear time, this is an offline preprocessing step that is performed only once, and subsequent learning tasks are conducted very efficiently. The effectiveness of the proposed method is empirically demonstrated using both synthetic and real-world data.

**Strengths:**

- This paper is clearly and systematically written, effectively utilizing figures and equations to concisely present the entire logical development of the paper.
- The methodology introduced with the Intersecting Communities Graph (ICG) offers a novel approach that is different from traditional graph machine learning methods. It provides a new pipeline that enables learning on large and dense graphs.
- By leveraging the Weak Graph Regularity Lemma, the paper provides a theoretically rigorous and efficient method for constructing ICG.
- The validity of the experimental results demonstrates the effectiveness of the proposed method.

**Weaknesses:**

- The proposed method has been shown to dramatically accelerate the learning of dense large-scale graphs. However, it is necessary to have a straightforward way to determine how dense a graph needs to be for the proposed method to be effective, or under what conditions of sparsity it remains effective. It seems that the conditions mentioned in lines 169-172, such as $K > N^4/E^2$, $K < d$, $E > N^{5/3}$, and $d > N^{2/3}$, might serve this purpose. However, these conditions are derived based on the Weak Regularity Lemma, and it is not clearly stated whether they can be used as practical criteria for the above judgments. Furthermore, in the experimental section, these indicators for each dataset do not appear to be clearly presented, making it difficult for readers to understand the existence of such effective criteria. By clearly demonstrating such criteria, the applicability of the proposed method could potentially be broadened.
- There are a few minor mistakes. For example, could (14) on line 110 be a mistake for (4)?

**Questions:**

- Is there a straightforward way to determine how dense a graph needs to be for the proposed method to be effective, or under what conditions of sparsity it remains effective? Are these the conditions mentioned in lines 169-172? To what extent are these criteria met in each dataset in the experimental section?

**Limitations:**

If there is no simple way to determine the applicability of the proposed method, it may limit the range of its practical application.

---

> ### Author Rebuttal · Authors · 2024-08-06
>
> We thank the reviewer for their comments and for highlighting the novelty of our approach and the overall quality of the paper. We address each of their questions/concerns below.
>
> > W1: …it is necessary to have a straightforward way to determine how dense a graph needs to be for the proposed method to be effective, or under what conditions of sparsity it remains effective…
>
> > Q1: Is there a straightforward way to determine how dense a graph needs to be for the proposed method to be effective…
>
> > L1: If there is no simple way to determine the applicability of the proposed method, it may limit the range of its practical application.
>
> **Answer:** We first note that after submitting the paper, we managed to significantly improve the asymptotics, which requires only a minor revision to the proof. **The multiplicative constunt in (6) is now $3N/2\sqrt{E}$ instead of $3N^2/2E$**. Moreover, we show that this asymptotic is essentially tight. Namely, there are graphs for which there does not exist any ICG that approximates them with an error less than $O(N/\sqrt{KE})$. This can be read from Theorem 1.2 in [1] (when converting their result to our definitions and notations). While this response is too short to show the proof, we are happy to send an anonymous PDF with the proof to the AC according to the rules and regulations of NerIPS, so you can verify it.   This improved theorem leads to a better comparison to MPNN methods. Now, ICG requires $K>N^2/E$ to guarantee that ICG approximates any given graph. For example, for a graph with $10^5$ nodes and average node degree $800$, one needs $K=125$ communities.
>
> This bound is still pessimistic for sparse graphs. As written in the paragraph in Line 173, we note that in practice, ICGs approximate also sparse graphs well under relevant structural assumptions. In the spare case where $K<N^2/E$, the theory only motivates the engineering, but does not guarantee a quantitive approximation bound for all lgraphs. In this case, we will write the following rule of thumb in the revised paper, which we actually observed and used in practice:
>
> "We observe in practice that when the relative Frobenius error between the graph and the ICG is less than 0.8, the cut norm error is usually small."
>
> Note that the Frobenius error is cheap to compute, and it is computed anyway as part of the optimization, while the cut norm, which is the actual theoretical target, is expensive to estimate.
>
> We will add this certificate to all of the tables in the paper.
>
> | METR-LA | PEMS-BAY | tolokers | squirrel | twitch-gamers |
> |---------|----------|----------|----------|---------------|
> | 0.44    | 0 .34    | 0.69     | 0.39     | 0 .8          |
>
> > W2: There are a few minor mistakes. For example, could (14) on line 110 be a mistake for (4)?
>
> **Answer:** Thank you for pointing that out. We will correct this error, along with any other minor typos.
>
> [1] Alon et al. Approximating sparse binary matrices in the cut-norm. Linear Algebra and its Applications, 2015.

---

> > ### Comment · Reviewer_2roR · 2024-08-12
> >
> > Thank you for your thorough response. My concerns appear to be largely resolved by the more precise mathematical descriptions you provided. Additionally, the supplementary experiments with real data have demonstrated the validity of your approach. Therefore, I will keep my rating unchanged.
> >
> > Thank you again.

---

> > > ### Author Response · Authors · 2024-08-12
> > >
> > > > Thank you for your thorough response. Additionally, the supplementary experiments with real data have demonstrated the validity of your approach.
> > >
> > > We thank the reviewer for acknowledging our rebuttal and participating in the discussion period.
> > >
> > > > My concerns appear to be largely resolved by the more precise mathematical descriptions you provided.
> > >
> > > We are glad to have resolved the reviewer's concerns. If the reviewer now sees the paper as worthy of publication, we hope they can increase their score above the threshold of acceptance to reflect that. We would like to use this opportunity to clarify remaining concerns that might prevent the a higher score.
> > >
> > > Regardless, we would like to warmly thank the reviewer again.

---

### Official Review · Reviewer_Z1J6 · 2024-07-14

**Soundness:** 3
**Presentation:** 3
**Contribution:** 3
**Rating:** 6
**Confidence:** 3

**Summary:**

The main idea of this paper is to break a graph into communities and approximate the graph with a limited number of these communities. The main idea of the work is rooted in the cut metric, which is defined for signals on the graphs and includes both graph structure and node features. They approximate the cut metric with a Frobenius norm version of the problem and prove a constructive version of the Weak Regularity lemma. Their reformulation allows the problem to be solved with a gradient descent algorithm. They also provide a version of subgraph gradient descent for larger graphs. They also theoretically analyze the approximation power of their subgraph gradient descent for finding the gradient on the whole graph. They introduce two models to be used after finding the communities and signals for these communities. The first model uses a Transformer architecture on the community signals, but the second one learns the community signals along the layers of their network. They provide time analysis on their work using random Erdős-Rényi graphs and show that their method is much more efficient than Graph Convolutional Network (GCN) in a forward pass of their model (used after finding the communities). They also get competitive results on node classification tasks for a few datasets and apply their work for a spatio-temporal task for traffic forecasting.

**Strengths:**

1. The paper has deep roots in the theory and does not neglect to provide theoretical analysis for any decision made in their process.

2. I am not an expert in the domain of signal processing for graphs, and I do not know to what extent the proofs in this paper are novel. However, they seem to present interesting results, and introducing them to the modern graph learning community is valuable in itself.

3. Their initial algorithm for finding the communities does not appear to be any simpler than learning on the graph by a Message Passing Neural Network (MPNN). However, it has the advantage of not requiring any hyperparameter tuning and only needing to be done once.

4. They have done a good job of finding spatiotemporal tasks where the graph structure is fixed, but the signals are changing. The initial community finding process load can be better justified in this setup.

**Weaknesses:**

1. Their analysis on the time complexity and efficiency of their method seems to be incorrect. Namely, in line 257, they mention that the message passing networks need $O(ED^2)$ operations. This is not correct. Message passing networks usually have a node-level processing which is $O(ND^2)$ and then a message aggregation process which is $O(ED)$, as there is no matrix multiplication along the edges and it is just adding the vectors of size D. Thus, the total complexity is $O(ND^2 + ED)$.
This is a significant problem as they mention their method is efficient if $K = O(Dd)$, where $D$ is the size of the features and $d$ is the average degree of the graph. By correct complexity of the message passing networks, we can see that their method only works comparably efficiently if $K = O(d)$, which is very small for most of the graphs. While Figure 2 suggests that their method is much faster than the GCN, the reason is that the graphs they sample are very dense, with $d = O(N)$ and $E = O(N^2)$. This is evident from Figure 6 in their appendix, where they use a more sparse graph with $d = 50$. We can see that when $K=100$, the time required by their method is similar to that of the GCN.

2. The paper gets unnecessarily complex and hard to understand, especially with the notations, which are excessive and sometimes inconsistent. For example, starting from Definition 2.1. the definition of the matrix norm is just used for the adjacency matrix A and feature matrix S in this paper, defining new matrices B and Z here just makes it confusing. At any point, if you require to define a formula that is general on all matrices, like when you describe the Frobenius norm, you can use names like X which does not feel to be a special matrix that might next definitions depend on it.

     Starting in Section 3.1, the use of C and P for finding the ICG on (A, S) using b vectors, and continuing with F at another point, makes it very confusing. One reason for this can be that B seems to be defined earlier in Definition 2.1, but it is not really connected to the b vectors used here.

      Definition 3.1 defines $\mathcal{Q}$ as a set of functions. However, starting with Definition 3.2, it uses the term "soft affiliation model" for $\mathcal{Q}$. I am not sure what is meant by "model" here and whether we are still referring to the same $\mathcal{Q}$. This is particularly confusing because, as I understand it, each member of $\mathcal{Q}$ is a function from $[N]$ to $\mathbb{R}$. Therefore, $\mathcal{Q}$ is a subset of $\mathbb{R}^N$, and the space defined for $[\mathcal{Q}]$ in Definition 3.2 is not clear to me.

3. While the main point is the efficiency of the method, the results for node classification are very limited, both in the number of datasets and the size of the datasets, and do not quite reflect the effectiveness of the method.

4. Their theory (Theorem 3.1) only works for very large K values for reasonably sparse graphs that can be seen in most datasets.

**Questions:**

1. Is there any time measured for fitting an ICG for a graph with the gradient descent? It seems to be at least as complex as learning the main task. However, I admit that it has the advantage of requiring no hyperparameter tuning and can be done once, with the condition that it can be done on large graphs with a reasonable calculation budget.

2. What is the relation of your work to graph coarsening methods? Methods such as DiffPool [1] seem to be learning a similar thing as a layer in the neural network, and thus their pooling is task-dependent. What are the advantages of your method over these techniques?

[1] Ying, Zhitao, et al. "Hierarchical graph representation learning with differentiable pooling." Advances in neural information processing systems 31 (2018).

**Limitations:**

The method seems to be more efficient for very dense graphs, which is discussed to some extent in the paper. However, there also seem to be some mistakes in their argument (discussed in the Weaknesses section).

---

> ### Author Rebuttal · Authors · 2024-08-06
>
> > W1: Their analysis on the time complexity…
>
> >L1: The method seems to be more efficient for very dense graphs…
>
> **Answer:** A general message passing layer applies a function on the concatenated pair of node features of each edge. We refer to this general formulation of MPNN, which takes $O(ED^2)$ operations for MLP message functions.
>
> The reviewer is correct that in some simplified message passing layers (e.g., GCN and GIN) the message is computed using just the feature of the transmitting node. Here, the complexity is reduced to $O(ED+ND^2)$. We will add this special case to the paper and compare asymptotics to ICGNN.
>
> Even for such simplified GNNs, for graphs with large average degree (the graphs of interest in our paper), taking $K$ less than $d$ makes sense. While standard benchmarks of large graphs are quite sparse (**tolokers**, $d=88$, **squirrel** $40$, **reddit** $50$), we believe that this is a result of the research community focusing on sparse graphs when developing methods for large graphs. However, in industrial applications, one often has to work with large and not very sparse graphs. See examples under **Choice of datasets** in the common response. For lack of such large and relatively dense publically available graphs, we have to make do with the available sparse large graphs, where our method performs remarkably well nevertheless. We hope the reviewer would agree that the lack of available high quality benchmarks should not be a reason to reject our contribution.
>
> Lastly, while taking $K=d$ may seem to give the same efficiency for GCN and ICG-NN, an ICG-NN has the advantage that it works with a regular array of a fixed dimens as the data structure. In contrast, GCN has to move between two data structures - the list of edges and the list of node features. This is less efficient in practice.
>
> > W2: The paper gets unnecessarily complex…
>
> > Starting in Section 3.1, the use of $C$ and $P$ for finding…
>
> **Answer:** We are happy to improve our notations in the revised paper.
>
> The matrix norms in our paper are not used on (non-negative) adjacency matrices but on (real-valued) differences between adjacency matrices. The distinction is critical. The cut norm of an adjacency matrix is just its L1 norm, but the cut norm of a general matrix is not. Thus, using the adjacency matrix $A$ to denote the variable inside the cut norm would be an abuse of notation. We will clarify this in the revised paper.
>
> We cannot replace the notation $P$ by $F$, since they are related by $P=QF$. We will change $b$ in (5) to $f$ to be consistent with the relation between $P$ and $F$. We will also change the generic matrix notation in Definition 2.1 to $M$.
>
> > Definition 3.1 defines $\mathcal{Q}$ as a set of functions...
>
> **Answer:** Definition 3.1. introduces rigorously the *soft affiliation model* $\mathcal{Q}$, which is a set of functions $q:[N] \rightarrow \mathbb{R}$. Given a soft affiliation model $\mathcal{Q}$, Definition 3.2. rigorously introduces *the soft rank-1 intersecting community graph model* $[\mathcal{Q}]$.
>
> We are happy for the opportunity to improve the wording and make the definition clearer. We will start Definition 3.2 with:
>
> "Let $d\in\mathbb{N}$, and let $\mathcal{Q}$ be a soft affiliation model. We define $[\mathcal{Q}]\subset\mathbb{R}^{N\times N}\times \mathbb{R}^{N\times D}$ to be the set of all elements..."
>
> > W3: While the main point is the efficiency of the method, the results for node classification...
>
> **Answer:** We conducted additional experiments using the **flickr** dataset, which is significantly larger than the datasets used in our initial submission. Our IGCNN with subgraph ICGm with 1% sampling rate, achieves competitive performance with competing methods that operate on the full graph.
>
> | Random          | DC-Graph        | GCOND           | SFGC            | GC-SNTK         | $\text{Sub-ICG}_u-\text{NN}$          |
> |------------------|-----------------|-----------------|-----------------|-----------------|----------------|
> | 44.6 ± 0.2       | 45.8 ± 0.1      | 47.1 ± 0.1      | 47.1 ± 0.1      | 46.6 ± 0.2      | **50.8** ± 0.1     |
>
> > W4: Their theory (Theorem 3.1) only works for very large K…
>
> **Answer:** We managed to significantly improve the asymptotics, which requires only a minor revision to the proof. **The multiplicative constant in (6) is now $3N/2\sqrt{E}$ instead of $3N^2/2E$**. Moreover, we show that this asymptotic is essentially tight.  Now, ICG requires $K>N^2/E$ to guarantee that ICG approximates any given graph. See **Improved asymptotic analysis** under the common response for more details.
>
> On the other hand, as written in the paragraph in Line 173, in practice, ICGs approximate also sparse graphs well. In the spare case the theory *only motivates* a class of graphs on which the computational method works, but does not guarantee a quantitative bound. This approach to motivating computational methods from theory is very common and accepted by the research community.
>
> > Q1: Is there any time measured for fitting an ICG…
>
> **Answer:** Run time analysis which shows a linear relationship between the runtime of ICG approximation and GCN be found in **Appendix F.4**, lines 1191-1194. The reduction in hyperparameter tuning is mentioned to motivate the usage of ICGs instead of MPNNs in **Section 8**, lines 381-382. We will highlight these points further in the final version.
>
> > Q2: What is the relation of your work to graph coarsening...
>
> **Answer:** In **Comparison of ICG-NN to graph coarsening** in the common response, we highlight the main differences. The summary is:
> 1. Coarsening methods do not have theoretical guarantees. ICGs provably approximate the graph.
> 2. Pooling methods do not asymptotically improve run-time. ICG-NNs do.
> 3. Pooling methods are MPNNs. ICG-NN is not interpreted as message passing.
> 4. Graph coarsening process representations on an iteratively coarsened graph. ICG-NNs process the fine information at every layer.

---

> > ### Comment · Reviewer_Z1J6 · 2024-08-11
> >
> > I thank the authors for their rebuttal and new experiments.
> >
> > > While standard benchmarks of large graphs are quite sparse, we believe that this is a result of the research community focusing on sparse graphs when developing methods for large graphs. However, in industrial applications, one often has to work with large and not very sparse graphs ...
> >
> > Thanks this has been an insightful response. I also think that the focus of most of the methods on large graphs has been on the sparse graphs and there is not enough work on denser graphs. I would also suggest adding this argument to your work and encouraging the community to work more on the denser, real-world graphs.
> >
> > **New experiments**: Thank you for running new experiments; the results are more convincing now. Just for the Reddit dataset, the average degree mentioned does not seem correct. It should be near 500 if you are using the original one, or near 100 if you are using the GraphSAINT version (according to the PyTorch Geometric library website). Also, ogbn-proteins can be a good candidate as it has a high average degree of near 600.
> >
> > **New Theory**: Thank you for updating; this is a more reasonable bound now.
> >
> > In general, I would like to encourage the authors to keep revising their paper as some parts are hard to follow in the current version and to update their manuscript with their rebuttal experiments and potentially more new datasets. Most of my concerns have been addressed, so I want to raise my score to 6.

---

> > > ### Author Response · Authors · 2024-08-11
> > >
> > > > Most of my concerns have been addressed, so I want to raise my score to 6.
> > >
> > > We warmly thank the reviewer for acknowledging our rebuttal and raising their score.
> > > > I would like to encourage the authors to keep revising their paper as some parts are hard to follow in the current version and to update their manuscript with their rebuttal experiments and potentially more new datasets.
> > >
> > > We will make sure to include all of the  points that were mentioned during the rebuttal in the final version of our paper.
> > > > It should be near 500 if you are using the original one, or near 100 if you are using the GraphSAINT version (according to the PyTorch Geometric library website).
> > >
> > > Thank you for pointing this out. You are correct, reddit's average degree is 491.99.

---

### Author Rebuttal · Authors · 2024-08-06

We thank the reviewers for their time and insightful comments. We respond to each concern in detail in our individual responses to the Reviewers, and provide here a summary of our rebuttal:

1. **Significance of contribution**: We want to stress that, while the numerical results are competitive and illustrate the merits of our proposed method, we believe that the theoretical results are the main contribution. **As far as we know, we are the first to bring the regularity lemma to the realm of practical numerical analysis**. We believe that this is the main focus of this paper. To emphasize the theoretical numerical analysis significance, we will add a review subsection about the regularity lemma and its computational aspects, which highlights the novelty of our paper (we already wrote it). We can copy this section as a comment to every reviewer that requests.

2. **Improved asymptotic analysis (Reviewer Z1J6 and 2roR)**: We managed to significantly improve the asymptotics of the ICG approximation bound in **Theorem 3.1**, which requires only a minor revision to the proof. **The multiplicative constant in (6) is now $\frac{3N}{2\sqrt{E}}$ instead of $\frac{3N^2}{2E}$**. Moreover, we show that this asymptotic is essentially tight. Namely, one can find a graph for which there does not exist any ICG that approximates it with an error less than $O(\frac{N}{\sqrt{KE}})$. This can be read off from **Theorem 1.2** in [1] (when converting their result to our definitions and notations). While this response is too short to show the proof, we are happy to send an anonymous PDF with the proof to the AC according to the rules and regulations of NeurIPS, so you can verify it.   This improved theorem leads to a better comparison to MPNN methods. Now, ICG requires $K>N^2/E$ to guarantee that ICG approximates any given graph.

3. **Additional experiments**:
    + **Comparison to graph coarsening methods (Reviewer hec5)** - We provide an empirical comparison in **Table 1** of **rebuttal.pdf** between our method and a variety of graph coarsening/summarization methods on the **reddit** and **flickr** datasets. ICG-NNs achieve state-of-the-art performance, further solidifying its effectiveness.
    + **Subgraph ICG-NN and ICG-NN on the large graph flickr (Reviewer Z1J6)** - To highlight the usage of subgraph sgd ICG-NN on large graphs, we conducted additional experiments using the **flickr** dataset, which is significantly larger than the datasets used in our initial submission. The results indicate that ICG-NNs with subgraph ICGm, using a 1% sampling rate, outperform competing methods that operate on the full graph.
    + **Run time analysis for the model's convergence (Reviewer hec5, XnC3)** - We measured the time (in seconds) until convergence for the GCN and $\text{ICG}_u-\text{NN}$ architectures across the node-classification tasks presented in **Appendix F.1, Table 2**. ICG-NNs converge significantly faster than GCN.
    + **Eigenvector initialization vs random initialization (Reviewer XnC3)** - We repeated the node classification experiments presented in **Table 2**, as described in **Appendix F.1**, using random initialization when optimizing the ICG, and compared the results to those obtained with eigenvector initialization. Eigenvector initialization improves performance.

4. **Choice of datasets (Reviewer hec5)**: We note that it is hard to find a dataset of large and non-sparse graphs. Currently, the community only focuses on sparse graphs in the context of large graphs when developing methods. As a result, researchers only publish sparse large datasets. We note that large graphs in real life are often not so sparse. For example, social networks can have an average degree of up to 1000, and transaction networks, user web activity monitoring networks, web traffic networks and Email networks can also be rather dense. Such networks, typically held by large companies, are not available to the public. Yet, knowing how to process large not-too-sparse graphs is of practical importance, and we believe that such graphs deserve a focus. As a first paper in this direction, it is hard to be competitive on datasets designed for sparse graphs. Still, our method performs remarkably well on large sparse graphs (in the paragraph in line 173  we give a possible explanation). We hope to see large dense real-life graphs open to the public in the future.

5. **Comparison of ICG-NN to graph coarsening (Reviewer Z1J6 and hec5)**: We highlight the main differences, which we will include in the final version of the paper.
   + Graph coarsening methods do not typically stem from theoretical guarantees, whereas ICGs provably approximate the graph.
   + GNNs with local pooling do not asymptotically improve run-time, as the first layer operates on the full graph, while our method operates solely on the efficient data structure.
   + In pooling approaches, the graph is partitioned into disjoint, or slightly overlapping communities, each community collapses to a node, and a standard message passing scheme is applied on this coarse graph. In our approach, each community is large, and occupies a significant portion of the graph, and different communities have a large overlap. In ICG-NNs, each operation on the community features has a global receptive field in the graph. Moreover, ICG-NNs are not of message-passing type: the (flattened) community feature vector $F$ is symmetryless and is operated upon by a general MLP in the ICG-NN, while MPNNs operate by the same function on all edges.
   + Graph coarsening methods process representations on an iteratively coarsened graph. ICG-NNs also process the fine node information at every layer.

We hope that our answers, along with the new experiments, address your concerns. We are looking forward to a fruitful discussion period.

[1] Alon et al. Approximating sparse binary matrices in the cut-norm. Linear Algebra and its Applications, 2015.

---

### Decision · Program_Chairs · 2024-09-25

**Decision:**

Accept (poster)

**Comment:**

The paper introduces the idea of Intersecting Communities Graph (ICG), which enables efficient learning on large non-sparse graphs. An offline preprocessing step (which runs in linear time in the number of edges) must be first used to construct the ICG, but after that approximate learning on the graph can be performed in time that is linear in the number of nodes (making it very useful in large dense graphs). The paper's theoretical contribution is significant, as it proves a constructive version of the Weak Regularity Lemma to efficiently construct an approximating ICG. Experiments also demonstrate the efficiency of the method across several datasets.

All reviewers seem to appreciate the technical contributions of the paper and the original ideas. While there were some questions from reviewers regarding practical use of the techniques (in which sparsity regimes it's more useful, how it compares with other large-scale graph learning techniques), reviewers seemed convinced by the rebuttals.